# OpenUnlearning: Accelerating LLM Unlearning via Unified Benchmarking of Methods and Metrics

**Vineeth Dorna**[*][†]    **Anmol Mekala**[*][†]    **Wenlong Zhao**[†]    **Andrew McCallum**[†]
**Zachary C. Lipton**[‡]    **J. Zico Kolter**[‡]    **Pratyush Maini**[‡][↑]

University of Massachusetts Amherst[†]    Carnegie Mellon University[‡]    DatologyAI[↑]
{vdorna,amekala}@umass.edu; pratyushmaini@cmu.edu

## Abstract

Robust unlearning is crucial for safely deploying large language models (LLMs) in environments where data privacy, model safety, and regulatory compliance must be ensured. Yet the task is inherently challenging, partly due to difficulties in reliably measuring whether unlearning has truly occurred. Moreover, fragmentation in current methodologies and inconsistent evaluation metrics hinder comparative analysis and reproducibility. To unify and accelerate research efforts, we introduce `OpenUnlearning`, a standardized and extensible framework designed explicitly for benchmarking both LLM unlearning methods and metrics. `OpenUnlearning` integrates 13 unlearning algorithms and 16 diverse evaluations across 3 leading benchmarks (TOFU, MUSE, and WMDP) and also enables analyses of forgetting behaviors across 450+ checkpoints we publicly release. Leveraging `OpenUnlearning`, we propose a novel meta-evaluation benchmark focused specifically on assessing the faithfulness and robustness of evaluation metrics themselves. We also benchmark diverse unlearning methods and provide a comparative analysis against an extensive evaluation suite. Overall, we establish a clear, community-driven pathway toward rigorous development in LLM unlearning research.

## 1 Introduction

LLMs often memorize sensitive, copyrighted or harmful content from their vast training data, raising privacy [6], safety [67] and legal [31, 61, 43] concerns. Ever increasing costs of pre-training and post-training [23, 54, 55] prevent re-training in response to deletion requests [36]. This has motivated the development of machine *unlearning* techniques that allow for "forgetting" training data via efficient post-training interventions [42, 36]. The goal of unlearning is to eliminate the undesirable influences from specific training data, while maintaining the overall behavior and performance.

There has been a recent surge in LLM unlearning research, yielding numerous proposed methods on several benchmarks. Modifying model weights to achieve unlearning is of the most interest, with many proposed approaches [76, 65, 33, 16, 40, 11, 29, 66, 17] . Concurrently, several benchmarks have been proposed to evaluate unlearning across a wide range of setups, covering aspects such as synthetic fine-grained unlearning, open-ended unlearning, knowledge, PII, memorization and privacy focused unlearning [39, 44, 46, 52, 33, 44, 57, 30, 14]. This volume of LLM unlearning research is marked by a notable fragmentation. Different benchmarks use different evaluations, with no consensus on the best evaluations and considerable criticism of existing evaluations [56, 48, 63, 77, 12, 38]. Evaluating unlearning is a nuanced task involving knowledge, privacy, and utility desiderata, which is arguably as hard as achieving unlearning itself [49, 37]. Unlearning research currently lacks a

---

[*]These authors contributed equally to this work.

unified, standardized framework, with current method implementations often tied to specific setups. This fragmentation limits the ability to rigorously evaluate the efficacy of unlearning methods across diverse settings. We envision LLM unlearning evolving within a shared framework that continuously integrates new and improved methods and evaluations—where unlearning methods iteratively improve on benchmarks, and evaluation metrics themselves improve through meta-evaluation and critical feedback. To catalyze this vision, we introduce `OpenUnlearning`: a unified and extensible benchmark designed to standardize, scale, and accelerate progress in machine unlearning for LLMs.

**A unifying framework.** We introduce `OpenUnlearning` as a one-stop repository for LLM unlearning, consolidating widely-used benchmarks, unlearning methods, evaluation metrics under different interventions. It is easy to use and extend, enabling the enrichment of benchmarks and a deeper analysis of unlearning algorithms. Through this standardized framework, we foster unified research efforts and expedite the creation of effective unlearning techniques and benchmarks.

**Evaluating evaluations.** Our framework moves the field towards a standardization of unlearning evaluations by conducting a meta-evaluation of unlearning metrics. To support this, we introduce a collection of over 450+ open-sourced models with known ground truth states, specifically designed to stress-test these metrics. This pool of models enables us to systematically compare 12 unlearning metrics against a set of desiderata that quantify their faithfulness (accuracy in detecting knowledge) and robustness (vulnerability to interventions). Together with corresponding meta-evaluation procedure, this forms the first benchmark of its kind for assessing and improving unlearning evaluation methods.

**Benchmarking unlearning techniques.** We compare 8 unlearning methods using a suite of 10 metrics, following Ramakrishna et al. [47]'s ranking procedure. While SimNPO [16] performs the best, we also note limitations with the ranking methodology. We release all the evaluated model checkpoints to encourage further community research into principled LLM unlearning benchmarking.

`OpenUnlearning` has been open-sourced[1] under the MIT license. Since its release in March 2025, it has already garnered wide attention in the LLM unlearning community, sitting at 250+ GitHub stars, 20k+ model downloads across 450+ publicly released checkpoints, and popular unlearning benchmarks[2] now also point to our repository as the official point of maintenance for their work.

## 2 Overview of LLM Unlearning

`OpenUnlearning` uses a common definition of LLM unlearning, where the goal is to eliminate the influence of "forget set" ($\mathcal{D}_{\text{forget}}$), from an LLM $f_{\text{target}}$ to remove associated model capabilities [36]. The process pursues two primary goals: (i) *Removal*, ensuring influence caused only by $\mathcal{D}_{\text{forget}}$ is substantially erased, and (ii) *Retention*, maintaining the LLM's utility on unrelated downstream tasks. The setup usually also involves a retain set disjoint from the forget set, used to aid and assess performance preservation.

Formally, given an original model $f_{\text{target}}$ trained on a dataset containing $\mathcal{D}_{\text{forget}}$, the unlearning process yields an unlearned model $f_{\text{unlearn}}$. The efficacy of unlearning is typically assessed using evaluation metrics, $M$, which quantify the remaining influence of $\mathcal{D}_{\text{forget}}$ on $f_{\text{unlearn}}$—e.g., by computing $M(f_{\text{unlearn}}, \mathcal{D}_{\text{forget}})$. Concurrently, utility metrics are used to measure the model's performance on general tasks and data outside of $\mathcal{D}_{\text{forget}}$, ensuring its overall capabilities are preserved.

**Unlearning methods:** Some LLM unlearning approaches are prompting-based, detecting sensitive queries at inference time and deploying obfuscation mechanisms [4, 41, 19]. But these are not practically scalable as forgetting results accumulate. Of greater interest is the removal of the forget set's influence directly from the weights. The techniques involved include finetuning with one or more of: (1) tailored loss functions [39, 16, 76, 11, 40], (2) optimization modifications [29, 66, 17], (3) localized parameter updates [33, 10, 20], and (4) alternative-data based approaches [40, 69, 7, 24, 39, 30].

**Benchmarks:** *Fine-grained unlearning* typically focuses on erasing influence of specific training instances from a forget set while preserving performance on related instances not present in the forget set. TOFU [39] introduces fine-grained *knowledge* unlearning using QA-style data from 200 fictitious

---

[1]Code ⌗: `github.com/locuslab/open-unlearning`; Models `huggingface.co/open-unlearning`
[2]TOFU [39] ⌗`github.com/locuslab/tofu`; MUSE [52] ⌗ `github.com/swj0419/muse_bench`

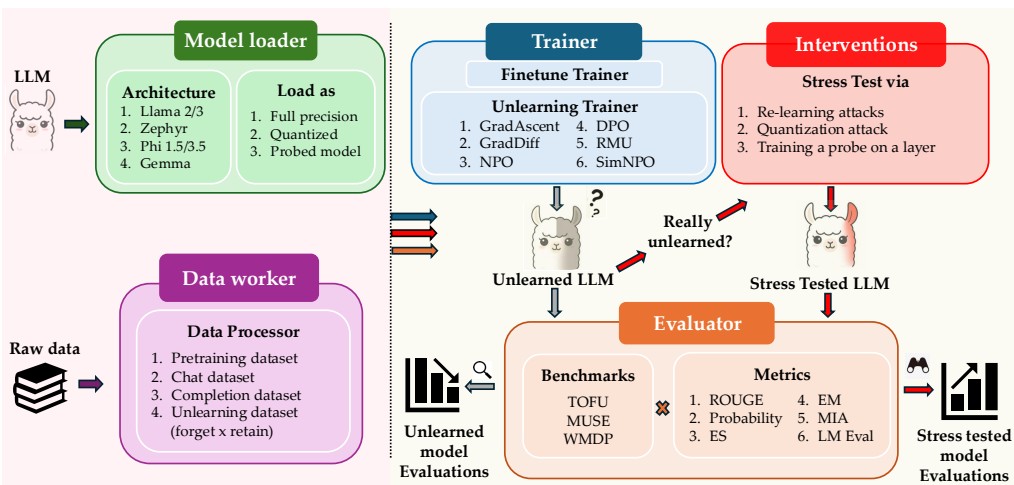

Figure 1: `OpenUnlearning` is an extensible library for benchmarking LLM unlearning methods and metrics. It provides a unified framework for implementing unlearning methods, unlearning metrics, and stress-testing tools to verify unlearning robustness. This figure illustrates the unlearning pipeline in terms of implementation-level components.

authors. KnowUndo [57] incorporates copyright and privacy aspects through datasets of books and synthetic author profiles. LUME [46] focuses on unlearning sensitive data from novels, biographies, and real-world figures. PISTOL [44] builds on TOFU with added structural relationships to study the effect of entity connectivity on knowledge unlearning. MUSE [52] also requires fine-grained unlearning, aiming to remove both knowledge, memorization and privacy influence of news articles and copyrighted books. *Open-ended unlearning* tasks do not target the removal of specific training data; instead, they aim to erase broader concepts or behaviors without access to a defined forget corpus. WMDP involves a safety-alignment focus, targeting which targets undesired behaviors from hazardous knowledge related to curated datasets [33]. RWKU [30] and *Who's Harry Potter* (WHP) task [14] require forgetting all knowledge related to famous entities. While benchmarks like TOFU, MUSE, PISTOL, LUME, and KnowUndo involve creating task models by injecting new knowledge via finetuning with the forget dataset; WMDP, RWKU and WHP [14] operate directly on off-the-shelf LLMs to remove existing influence.

**Unlearning evaluations:** Each benchmark task involves multiple evaluations metrics that judge for unlearning success and for general utility preservation. These range from simple probability judgements in TOFU, to MIA-attack based metrics in MUSE, with dozens of metrics across benchmarks in the literature. Evaluating unlearning success is difficult, with several subsequent works questioning the reliability of benchmark metrics in various aspects [32, 38, 62, 12, 77].

## 3 OpenUnlearning

The significant volume of research in LLM unlearning lacks unification both in technical implementations and in both unlearning method implementation and unlearning evaluation methodology. Existing benchmarks are implemented with a structure that makes it difficult to integrate with newer ones, hindering their adoption, and creating barriers to reproducibility that slow down progress. Moreover, unlearning methods and evaluation metrics aren't consistently extended across benchmarks, preventing standardization and comprehensive comparative analysis. We give a few examples of this fragmentation that cover key parts of the unlearning pipeline, from unlearning algorithms, to data processing, and evaluations,

1. **Fragmented evaluations of methods:** New methods are not implemented in all benchmarks. For example: UNDIAL [11] is not implemented on any of TOFU, MUSE and WMDP; NPO [76] is implemented with a different formulation for TOFU v/s MUSE; RMU was introduced only for WMDP etc. Similarly, evaluation metrics like MIA from MUSE [52] are not implemented in TOFU; and LM Eval Harness benchmarks used in WMDP can be extended to TOFU, MUSE.

Table 1: Overview of existing `OpenUnlearning` components and their available feature variants. The design is easily extensible, allowing users to seamlessly contribute new features.

| Component | | Variants |
|---|---|---|
| **Models** | | LLAMA-2, 3.1, 3.2 [59, 23]   ZEPHYR-7B [60]   PHI-1.5, 3.5 [34, 1] |
| | | QWEN-2.5 [45]   GEMMA [22] |
| **Unlearning algorithms** | | GradAscent, GradDiff, IdkDPO, IdkNLL [39]   NPO [76]   SimNPO [16] |
| | | RMU [33]   UNDIAL [11]   AltPO [40]   CE-U [71]   PDU [15] |
| | | WGA [64]   SatImp [73] |
| **Datasets** | | TOFU: bios [39]   WMDP: cyber, bio [33]   MUSE: news, books [52] |
| **Evaluation suites** | | TOFU [39]   MUSE [52]   WMDP [33]   LM Eval [21] |
| **Metrics** | **Mem.** | Verbatim Prob. / ROUGE [39, 52]   Knowledge QA- ROUGE [39, 52] |
| | | Extraction Strength [5]   Exact Memorization [58] |
| | **Privacy** | Forget Quality [39]   LOSS [74]   ZLib [5]   GradNorm [62] |
| | | MinK [51]   MinK++ [75]   Privacy Leakage [52] |
| | **Utility** | Truth Ratio, Model Utility [39]   LM-Eval [21] (WMDP, MMLU, etc.) |
| | | Fluency [40] |
| **Stress tests** | | Relearning [27, 38, 37, 63]   Quantization [77]   Probing [38, 50, 63] |

2. **Disparate implementations of core components:** Several approaches involve customized loss functions [76, 39, 16, 11] and others make adjustments to optimization steps [66, 29, 17]. These techniques could be modularized and reused across tasks for deeper investigation and a fair comparison. Evaluation metrics use many common functionalities which can be shared across metric implementations (eg. probability, ROUGE-score and MIA statistics). Dataset pre-processing is separately implemented across datasets and benchmarks, while there are many common data types: like the pre-training corpora in WMDP and MUSE, and chat-style prompts in TOFU and RWKU. Some works have proposed stress tests for assessing the robustness of unlearning which could easily be a common feature across benchmarks.

To address this, we introduce `OpenUnlearning`: a unified, extensible pipeline that consolidates benchmarks, methods, evaluation metrics, datasets, and stress-tests under one roof (see Figure 1) to streamline unlearning implementations, benchmarking, and accelerate research.

## 3.1   Design of OpenUnlearning

Figure 1 gives an overview of `OpenUnlearning`'s components. Our framework is designed with ease-of-use and easy extensibility in mind. All features are implemented in a structured, modular fashion, simplifying the process for researchers to integrate new datasets, evaluation metrics, unlearning methods, and entire benchmarks. Hydra [70] is used for configuration management, with `YAML` files specifying each pipeline component and experiment parameters. This helps users effortlessly swap in modules and easily launch an experiment with a single command. A variety of modules, including model-loaders, trainers, dataset preprocessors, evaluation suites, evaluation metrics, experiment types and stress-test interventions are joined together in `OpenUnlearning` (listed in Table 1).

## 3.2   Design of modules

The procedure of extending `OpenUnlearning` with a new module variant generally involves two simple steps. (1) **Create and register a handler.** The `Python` class or function encapsulating the component's logic is implemented then registered to be accessed via a string key. (2) **Create the**

**(a) Method implementation leveraging HuggingFace Trainer, followed by registration.**

```python
from transformers import Trainer
class Unlearner(Trainer):
  def compute_loss(self, ...):
    ...
  def get_optimizer_cls_and_kwargs(...):
    # custom optimizer
    ...
  def _inner_training_loop(self, ...):
    # modify training logic
    ...
_register_trainer(Unlearner)
```

**(b) Configuration: create a YAML config specifying Training args and method parameters.**

```yaml
handler: Unlearner # map registered name

args: # HuggingFace Trainer args
  num_epochs: 10
  learning_rate: 1e-5
  optim: shampoo

method_args:
  alpha: 1.0
  switch_every_n: 10
  retain_loss_type: NLL
```

Figure 2: Illustration of implementing a hypothetical unlearning method in `OpenUnlearning`

**config.** The configuration `YAML` file names the handler key and specifies its parameters. Figure 2 provides an example illustrating this procedure for a new unlearning method.

**Features:** We currently support 13 unlearning algorithms, 8 model architectures, and 5 datasets ranging from chat to pretraining. Among existing benchmarks, we focus on the three most cited and used TOFU [39], MUSE [52], WMDP [33] benchmarks. The framework includes a diverse set of metrics to assess model performance, including 16 unlearning metrics from existing benchmarks, as well as additional evaluations by integrating LM Eval Harness [21]. We also support three stress-testing approaches, which are essential for testing the robustness of unlearning, usually critical for model-owners in verifying compliance. All these features are summarized in Table 1 by component and variant. Our integration enriches each benchmark by enabling the use of metrics originally developed for others. For example, PrivLeak, initially introduced in MUSE, is now available in TOFU. More details on these technical benchmark improvements can be found in Appendix C.1. We also encourage community contributions by providing detailed guidelines for adding new benchmarks, unlearning methods, and evaluation metrics. This has already resulted in contributions from the community, with implementations for works like [11, 66, 72].

`OpenUnlearning` is a living framework, and our design choices are built keeping easy integration of new components in mind. For instance, since the public release of our repository (with just TOFU and MUSE benchmarks) we introduced the WMDP benchmark, unlearning methods like RMU [33], UNDIAL [11], AltPO [40]; evaluations like ES [5], EM [58], MIA [13] and integrated evaluations like MUSE's PrivLeak (into TOFU) and LM Eval Harness [21] (to enable WMDP evaluation) among many others. Additionally, we encourage community contributions by providing detailed guidelines for adding new benchmarks, unlearning methods, and evaluation metrics. This has already resulted in contributions from the community, with implementations for works like [11, 66, 72]. Currently, each module supports several variants, with 3 popular LLM unlearning benchmarks, 5 task datasets, 13 unlearning methods, 16 evaluation metrics, 8 LLM architectures and 3 stress-tests.

## 4 Evaluating Unlearning Evaluations

Reliable evaluations for unlearning are essential for regulatory compliance and data privacy, yet remain challenging [32, 49, 37], especially for LLMs, due to ambiguity between memorization and generalization. We propose two minimal necessary desiderata—*Faithfulness* and *Robustness*—guided by our meta-evaluation framework, to promote trustworthy unlearning metrics (Figure 3).

Our meta-evaluation uses a test-bed of models with known ground truths to objectively assess metrics. We employ the TOFU benchmark [39] with the improvements described from Appendix C.1 with the `forget10` unlearning task (forgetting 10% of TOFU) comprising 400 examples. We use the LLAMA-3.2 1B model [23], analyzing 12 unlearning metrics adjusted to $[0, 1]$ scale (see Appendix C.1). While the TOFU benchmark setup we choose makes simplifying assumptions about unlearning data distribution and target model behavior, such a synthetic setup enables controlled evaluation of metric properties that would be difficult to assess systematically with purely real-world data. With this approach, we are able to establish a minimal set of properties that any reliable unlearning evaluation metric should satisfy.

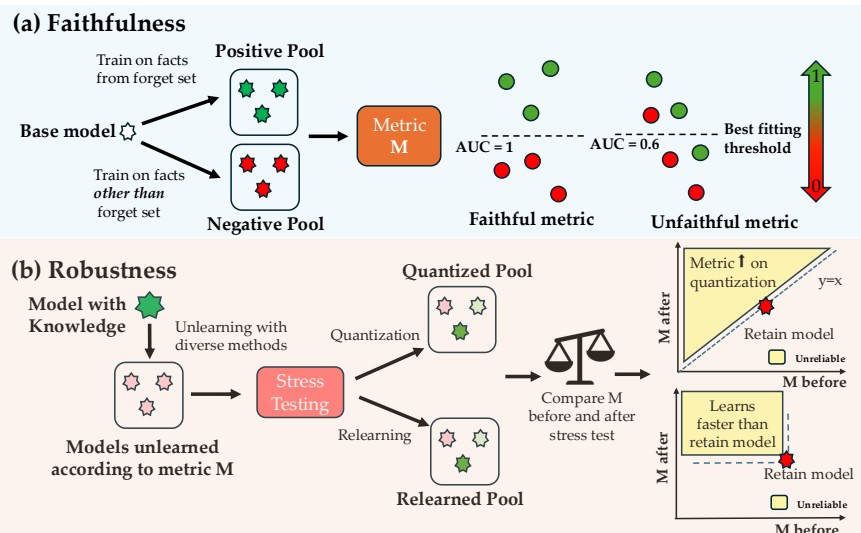

Figure 3: **Meta-evaluation of unlearning metrics**: (1) *Faithfulness*: the metric distinguishes models with and without target knowledge, reflected by high AUC; (2) *Robustness*: the metric value does not increase under benign changes (e.g., quantization) and does not improve faster than a retain model under non-benign changes (e.g., relearning).

## 4.1 Faithfulness

> **Faithfulness**
>
> **Motivation.** Unlearning evaluations may not faithfully reflect an LLM's knowledge.
> **Desideratum.** A faithful metric accurately reflects the presence of targeted knowledge by assigning consistently higher scores to models possessing it than to those lacking it.

LLMs often fail to regurgitate facts that remain encoded in their parameters when prompted, making it hard to tell whether a model truly forgot a target fact or simply refrained from exposing it [12, 38, 48, 63]. For example, work by Doshi and Stickland [12] shows that simple paraphrasing of inputs can yield a tenfold increase in evaluation scores on 'unlearned' models, indicating that the apparent forgetting may only be superficial. "Deeper" evaluation metrics aim to quantify this knowledge more faithfully, like Truth Ratio [39], GCG [18], or by using prompt engineering [63, 53, 56].

On the other hand, evaluation metrics can register misleadingly high scores without the presence of the target knowledge [39]. For example, in a question-answering evaluation using a simple ROUGE score, a model might achieve a high score by matching the parts of the target unrelated to the target fact. This calls for metrics that are **faithful** to the knowledge encoded in the model weights.

We measure *faithfulness* as the ability of metrics to distinguish between models trained with the forget dataset's knowledge (the *positive pool*, $P$) and those trained without it (the *negative pool*, $N$): (i) Each pool has 30 diverse models trained under varying conditions. (ii) These variants present the target `forget10` information for pool P models in diverse, challenging formats (e.g., biography vs. QA, paraphrases). Pool N models serve as negative controls, using similarly structured data lacking this target information using various perturbations and alternative datasets. (iii) Metric scores yield two distributions: $m(P)$, $m(N)$ (for $P$ and $N$), and we compute AUC-ROC to quantify their separability. (iv) We select a classification threshold optimizing accuracy, which is subsequently used in robustness tests.

$$\text{Faithfulness} = \text{AUC-ROC}(m(P), m(N)) \qquad (1)$$

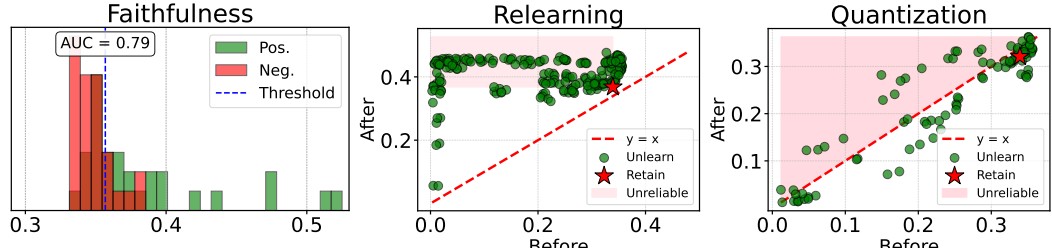

Figure 4: For the ROUGE metric we evaluate faithfulness (left) and robustness to quantization (middle), and relearning (right). Faithfulness achieves an AUC of 0.79, indicating substantial prediction overlap between models trained with and without the target knowledge. Relearning robustness is 0.48, showing many unlearned models re-acquire knowledge faster than the retain model upon re-exposure. Quantization robustness is 0.93, reflecting no distinctive trend of metric spikes post-quantization.

## 4.2 Robustness

> **Robustness**
>
> **Motivation.** Unlearning evaluations can be vulnerable to stress-testing interventions.
> **Desideratum.** A robust metric's positive assessment of unlearning should (1) not flip upon benign model interventions; and (2) behave comparably to a model truly unfamiliar with the data under non-benign interventions.

Robustness of unlearning metrics is probed using various stress-test interventions. These include (1) relearning attempts, where the unlearned model is further trained to potentially recover the forgotten information [38, 27, 37, 63]; (2) information extraction via manipulating the model's internal representations [3, 38, 50, 63, 2]; and (3) applying techniques like quantization [77]. Benign interventions, such as model quantization or relearning on non-forget data, do not reintroduce the forgotten knowledge. In contrast, non-benign interventions—like relearning directly on the forget set—explicitly re-expose the model to the targeted data. These stress tests have revealed that several unlearning evaluation metrics may be unreliable, often signaling successful unlearning even when the underlying knowledge remains recoverable.

For example, Zhang et al. [77] show that the PrivLeak metric [52] that previously reported a model as successfully unlearned can effectively 'flip' after a benign intervention, revealing that the targeted knowledge was perhaps never truly erased [77]. Such significant fluctuations under stress tests undermine the reliability of evaluation metrics. Furthermore, models unlearned with respect to a metric can exhibit high susceptibility on metric evaluation to non-benign interventions like relearning, where evaluation metrics show an unusually rapid return of the supposedly forgotten knowledge even with minimal retraining effort [17, 16]. Robustness assesses stability under interventions such as relearning, probing and quantization. While probing was previously used by Wang et al. [63], Seyitoğlu et al. [50], Lynch et al. [38] to stress-test unlearning, in our setup, we found that probed models perform very poorly, with low scores across all metrics and show little discernible trends. Some probing results are shown in Appendix E.3.

**Robustness to Relearning:** We evaluate metric scores before ($m^a$) and after ($m^b$) relearning on forget-set data. Then, we compare relative metric score recovery rates between unlearned ($m_{\text{unl}}$) and retain ($m_{\text{ret}}$) models, where higher $R$ implies greater robustness.

$$r = \frac{m_{\text{ret}}^a - m_{\text{ret}}^b}{m_{\text{unl}}^a - m_{\text{unl}}^b}, \quad R = \min(r, 1). \tag{2}$$

**Robustness to Quantization:** We quantize models to 4-bit precision and compute scores before and after quantization, where higher $Q$ implies greater robustness.

$$q = \frac{m_{\text{unl}}^b}{m_{\text{unl}}^a}, \quad Q = \min(q, 1). \tag{3}$$

Table 2: Meta-evaluation of 12 unlearning metrics for Faithfulness and Robustness. Robustness is assessed using two stress-testing methods: quantization and relearning, with their harmonic mean reported as Agg. An overall aggregation across both Faithfulness and Robustness is reported in the first Agg. column. Higher scores indicate better performance (↑) in all dimensions. The best values are shown in bold, and the second-best values are underlined.

| Metrics | Agg. ↑ | Faithful. ↑ | Robustness ↑ | | |
| --- | --- | --- | --- | --- | --- |
| | | | Agg. ↑ | Quant. ↑ | Relearn ↑ |
| Extraction Strength | **0.85** | 0.92 | **0.79** | **0.95** | 0.68 |
| Exact Mem. | 0.80 | 0.90 | 0.72 | 0.92 | 0.59 |
| Truth Ratio | 0.73 | **0.95** | 0.59 | 0.92 | 0.43 |
| Para. Prob. | 0.73 | 0.71 | 0.75 | 0.60 | **0.98** |
| Para. ROUGE | 0.72 | 0.89 | 0.61 | 0.93 | 0.45 |
| Probability | 0.72 | 0.82 | 0.65 | 0.60 | 0.70 |
| ROUGE | 0.70 | 0.79 | 0.64 | 0.93 | 0.48 |
| Jailbreak ROUGE | 0.69 | 0.83 | 0.59 | 0.85 | 0.45 |
| MIA - ZLib | 0.71 | 0.92 | 0.57 | 0.56 | 0.59 |
| MIA - MinK | 0.67 | 0.93 | 0.52 | 0.48 | 0.57 |
| MIA - LOSS | 0.66 | 0.93 | 0.52 | 0.48 | 0.57 |
| MIA - MinK++ | 0.61 | 0.81 | 0.48 | 0.61 | 0.40 |

### 4.2.1 Realistic Model Filtering

We enforce practical constraints by filtering models with: (i) Utility drops exceeding 20%. (ii) Insufficient unlearning w.r.t. the considered metric (more than the threshold computed in §4.1's *faithfulness* analysis). Models which exhibit substantial model utility drops are unusable in practice and thus unlikely to inform robustness. Additionally, models that aren't unlearned w.r.t a metric are uninteresting for robustness analysis, since they do not reflect realistic scenarios where some kind of unlearning is observed before models are stress tested. The case of interest is when an ostensibly performant LLM exhibits low scores according to a chosen metric, indicating unlearning, and practitioners require confidence in the metric's judgement.

We analyze roughly 400 diverse models from various unlearning methods to reflect realistic use cases. We ensure diversity by using models unlearned using the GradDiff, IdkDPO, IdkNLL [39], NPO [76], SimNPO [16], AltPO [40], UNDIAL [11] and RMU [33] unlearning methods (methods described in Appendix §C.5 and hyperparameters in §F.2). This aligns the distributions between the unlearned model pools used in our analysis and unlearned models selected by practitioners.

### 4.3 Aggregation of Metrics

We consolidate evaluations through harmonic mean, ensuring balanced performance across criteria:

$$\text{Robustness} = \text{HM}(R, Q), \quad \text{Overall} = \text{HM}(\text{Faithfulness}, \text{Robustness}) \tag{4}$$

An effective unlearning metric must be both faithful in representing unlearning and robust in its measurements; a trivial constant-value metric, for instance, would be robust but entirely unfaithful. To holistically assess a metric, we aggregate these distinct qualities using the Harmonic Mean (HM), as this ensures that a high final score demands strong performance in all constituent parts. Figure 4 illustrates these distributions and scores for the ROUGE metric as an example. Further methodological considerations, including comparisons to prior work, are detailed in Appendix E.4.

### 4.4 Results and Discussion

Table 2 highlights key insights: (i) **Extraction Strength (ES)** [5] emerges as most reliable overall, aligning with Wang et al. [63]. (ii) **Truth Ratio** has superior faithfulness but lower robustness, ranking third overall. (iii) Metrics based on raw probabilities or ROUGE scores have moderate faithfulness and robustness, limiting their reliability. (iv) Membership inference (MIA)-based metrics demonstrate high faithfulness but lack robustness, cautioning against relying solely on MIA metrics for assessing unlearning. This sensitivity raises concerns about the reliability of the MIA-based privacy assessments

Table 3: Comparison of unlearning methods on the TOFU task, showing overall aggregate (Agg.), memorization (Mem.), privacy (Priv.), and utility (Utility) scores. Higher scores indicate better performance ($\uparrow$). Initial finetuned is the target model before unlearning and Retain model is the gold standard target model. The best values are shown in bold, and the second-best values are underlined.

| Method | Agg. $\uparrow$ | Mem. $\uparrow$ | Priv. $\uparrow$ | Utility $\uparrow$ |
|---|---|---|---|---|
| Init. finetuned | 0.00 | 0.00 | 0.10 | 1.00 |
| Retain | 0.58 | 0.31 | 1.00 | 0.99 |
| SimNPO [16] | **0.53** | 0.32 | **0.63** | **1.00** |
| RMU [33] | 0.52 | 0.47 | 0.50 | 0.61 |
| UNDIAL [11] | 0.42 | 0.27 | 0.48 | 0.78 |
| AltPO [40] | 0.15 | 0.63 | 0.06 | 0.95 |
| IdkNLL [39] | 0.15 | 0.08 | 0.17 | 0.93 |
| NPO [76] | 0.15 | 0.52 | 0.06 | 0.99 |
| IdkDPO [39] | 0.14 | 0.56 | 0.06 | 0.95 |
| GradDiff [39] | 9e-3 | **0.97** | 3e-3 | 0.79 |

in unlearning contexts as introduced by Shi et al. [52], as even benign interventions can reverse unlearning effects, as observed in Zhang et al. [77].

Our extensive model testbed supports ongoing development of improved, practical unlearning metrics. Our testbed comprising 450+ models — including those from pools $P$, $N$, and various unlearned model checkpoints — offers a valuable platform for the creation and rigorous assessment of improved unlearning evaluation metrics. Metrics validated on this testbed can then be applied with greater confidence to real-world unlearning scenarios. Our overarching goal is to stimulate the development of more faithful and trustworthy metrics, leveraging the insights from our meta-evaluation framework. This meta-evaluation setup can be expanded by incorporating more diverse unlearning setups, model architectures and newer methods. Newer adversarial model setups will be needed to challenge metrics as they improve on existing testbeds. Such a dynamic approach ensures that unlearning methods and their meta-evaluations can mutually inform each other, driving progress as unlearning research advances.

## 5 Benchmarking Unlearning Methods

Unlike prior works with limited baselines and metrics, `OpenUnlearning` provides a standardized and scalable framework to conduct a large-scale comparison of various unlearning methods. We demonstrate this by evaluating 8 unlearning methods using 10 evaluation metrics on TOFU.

**Unlearning methods:** `OpenUnlearning` enables evaluation across a broader range of methods, including SimNPO [16], RMU [33], AltPO [40], NPO [76], UNDIAL [11], as well as baselines like IdkPO, IdkNLL, and GradDiff [39]. See Appendix C.5 for each method's definition.

**Evaluation metrics:** We evaluate unlearning methods using memorization metrics validated in our meta-analysis, alongside privacy and utility metrics. Using the TOFU benchmark, and following the SemEval 2025 LLM Unlearning Challenge's ranking procedure [47], we compute a composite score by aggregating metrics from the three categories: memorization (using the 4 top-performing knowledge metrics from §4's metric meta-evaluation: ES, EM, Truth Ratio, Paraphrased Probability), privacy (4 MIA metrics), and utility (2 metrics, including TOFU's Model Utility and forget-set fluency). Exact details of our metric aggregation are in Appendix F.1. Note that the memorization score (reported in Table 3) corresponds to forgetting: higher Mem. indicates less knowledge.

**Tuning strategy:** To ensure fairness, 27 hyperparameter tuning trials are allocated per method, as tuning can significantly improve performance of even simple baselines [63]. Due to the impracticality of tuning on privacy metrics, that require the presence of i.i.d. holdout datasets and oracle retain models (i.e., models trained solely on the retain set, with no exposure to the forget set), we validate models only on accessible metrics that capture memorization and utility. Additionally, model selection during tuning can significantly affect rankings (Appendix F.2).

**Results and discussion:** While memorization, privacy, and utility each capture a distinct aspect of unlearning quality, aggregating them using a harmonic mean (Table 3), results in SimNPO [16] ranking first. Although its memorization score trails that of others, it remains close to the retain model's level, avoiding over-unlearning. SimNPO fully preserves utility and achieves competitive privacy results, striking a balance across all three criteria. The next best performer is RMU, which demonstrates strong memorization and privacy but suffers a significant drop in utility.

Here, we note a tradeoff between reducing memorization and improving data privacy during the unlearning process. Memorization evaluation penalizes high likelihood on forget data; while privacy metrics penalize both unusually high and low likelihoods. Thus, methods that under-unlearn (e.g. IdkNLL, which yields a low memorization score i.e. less forgetting) score lower on privacy. On the other hand, methods like GradDiff over-unlearn, forgetting too aggressively, yielding a high memorization score. This leads to poor privacy performance, as the model's behavior deviates significantly from that of the retain model. This suggests that detecting and halting unlearning once the model's behavior has reverted to its "default" state is crucial to ensure privacy.

Because different ranking schemes can produce very different rankings (Table 3 v/s Appendix Table 6), it is critical to choose an appropriate method ranking procedure and aggregate metrics. Additionally, there is a lack of standardization on which metrics are suitable for model selection versus final evaluation (elaborated upon in Appendix F.2). While identifying the ideal ranking method and model selection approach is beyond our scope, we release all unlearned model checkpoints from our study to support future research on fair evaluation.

## 6 Conclusion

The field of LLM unlearning has faced challenges due to fragmented methodologies and inconsistent evaluations. To address this, we introduced `OpenUnlearning`, a standardized and extensible framework that unifies research efforts by integrating 13 unlearning algorithms, 16 evaluation metrics, and 3 major benchmarks. This comprehensive platform enabled us to conduct a novel meta-evaluation of unlearning metrics, assessing their faithfulness and robustness, and to perform large-scale benchmarking of unlearning methods. Our meta-evaluation identified Extraction Strength (ES) and Exact Memorization (EM) as particularly reliable metrics, with Truth Ratio also showing high faithfulness. Benchmarking revealed SimNPO and RMU as strong performers, though we also observed significant sensitivities in ranking. At the same time `OpenUnlearning`, by providing a common ground and releasing numerous model checkpoints, establishes a clear pathway for the community towards more rigorous, reproducible, and accelerated development of robust unlearning techniques and evaluation protocols, ultimately fostering safer AI deployments.

## 7 Acknowledgments

We thank all contributors for adding new unlearning methods and metrics. We also appreciate their continued support through active use of the repository and valuable feedback that helps improve the codebase. We also acknowledge the IESL lab at University of Massachusetts Amherst for providing compute resources for this work. PM is supported by funding from the DARPA GARD program and OpenAI's Cybersecurity Grant Program.

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

# Appendix

## A    Limitations

We also note some limitations of our framework and analysis. Firstly, it is limited by the existing popular benchmarks its supports, which have been regarded as "weak measures of unlearning progress" [56]. The setups may not accurately reflect realistic model learning or unlearning dynamics, with the underlying forget-retain paradigm itself warranting further scrutiny [56]. There's a clear need for more realistic, yet controlled, fine-grained unlearning benchmark setups beyond the currently popular benchmarks. Secondly, while our meta-evaluation of metrics and comparison of methods is a valuable step, its findings need to be extended to more unlearning setups and unlearning algorithms, to gain a greater understanding of the best and comprehensive ways to quantify unlearning. Finally, while our meta-evaluation focuses on knowledge faithfulness and metric robustness as minimal desiderata, these might not be a comprehensive set of desiderata for good unlearning metrics.

## B    Broader Impact

The widespread deployment of AI systems in domains ranging from conversational assistants and recommendation systems to self-driving vehicles and medical diagnostics raises important concerns about privacy, safety, and regulatory compliance. As these systems are deeply integrated within society, the ability to remove unwanted or sensitive information from deployed models ("unlearning") is essential to maintain safety, reliability and uphold legal requirements.

Our work on a unified, extensible LLM unlearning benchmark accelerates progress toward reliable, scalable unlearning solutions. By standardizing implementations of unlearning methods, evaluation metrics, and stress tests across diverse tasks and datasets, we lower the barrier for both academic and industrial adoption. This facilitates rapid iteration on novel techniques, ensures consistent measurement of privacy and utility trade-offs, and enables model governance workflows that can respond promptly to deletion or correction requests.

In the long run, advances enabled by this framework will support trustworthy AI deployment in safety-critical and highly regulated settings. From ensuring that autonomous vehicles do not retain outdated or hazardous driving data, to empowering personalized assistants with user-controlled memory, robust unlearning mechanisms will be a cornerstone of ethical, privacy-preserving machine learning. By fostering community collaboration and transparent evaluation, our research paves the way for AI systems that adapt responsibly to evolving societal norms and regulatory landscapes.

## C    Additional details on OpenUnlearning's components

### C.1    Unlearning benchmarks

**TOFU:** A synthetic fine-grained knowledge-unlearning benchmark with 200 fictitious author profiles, each offering 20 QA pairs and a defined "forget set", and a finetuned chat LLM. TOFU's primary metric is Truth Ratio, which measures the *relative* likelihood of the true answer after unlearning.

**MUSE:** A memorization and knowledge unlearning benchmark targeting the removal of books and news articles from a finetuned LLM. MUSE evaluates for memorization (via verbatim reproduction rates), knowledge (via question-answers) and privacy protection (using membership inference attacks).

**WMDP:** An alignment-focused benchmark of 3,668 multiple-choice questions probing hazardous knowledge in biosecurity, cybersecurity, and chemical security, paired with corresponding unlearning corpora and off-the-shelf chat LLMs. WMDP assesses a model's ability to forget dangerous capabilities while preserving general performance.

**Improvements:** In evaluations, TOFU reuses training questions, raising concerns about overfitting and inflated metrics. To mitigate this, we evaluate on paraphrased questions in our meta-evaluation and benchmarking. We also extend TOFU with privacy-based metrics from MUSE via PrivLeak [52] and introduce additional MIA attacks. For this we create new holdout datasets by replicating the

original TOFU data generation setup.[3] We add MIA beyond Min-K [51] to MUSE. Given the poor quality and tokenization issues users faced with the PHI-1.5 and LLAMA-2 models from TOFU, we introduce new starter target models. OpenUnlearning provides three sizes of the recent LLAMA-3 models: 1B, 3B, and 8B, giving users greater flexibility to experiment. Additionally, we augment both TOFU and MUSE with metrics such as Extraction Strength [5], Exact Memorization [58], and Forget Fluency [40]. We integrate OpenUnlearning with LM Eval Harness [21] to assess general LLM capabilities that identify post-unlearning degradations, in addition to enabling WMDP evaluations. Several contemporary works can further enhance these benchmarks. We plan to continuously improve the framework by adding-and encouraging contributions of-new features and metrics to both existing and future benchmarks, such as the recent work by Thaker et al. [56].

## C.2   Datasets

In machine unlearning, benchmarks typically structure data into two primary components: (1) forget sets, which contain text corpora and queries designed to test whether the model has successfully erased targeted information, and (2) retain sets, which verify that the model preserves unrelated, desirable knowledge. Beyond this fundamental split, unlearning benchmarks often include additional variations to test algorithmic robustness. For example, scaling splits vary the size of the forget set to assess how well algorithms handle larger deletion requests, while topic-based splits examine whether forgetting specific content impacts retention across semantically related or unrelated domains [39, 52]. These nuanced splits are essential for assessing scalability, generalization, and sustainability of unlearning methods under realistic conditions.

|                   (a) Dataset Handler                   |              (b) Dataset Configuration               |

```
class PretrainingDataset(Dataset):         MUSE_forget:
    def __init__(self, hf_args, ...):        handler: PretrainingDataset
        ...                                   args:
                                                hf_args:
    def __getitem__(self, idx):                   path: "muse-bench/MUSE-News"
        ...                                       name: "raw"
        return item                               split: "forget"
                                                text_key: "text"
_register_data(PretrainingDataset)          max_length: 2048
```

Figure 5: Adding a dataset in OpenUnlearning: (a) the Python handler class implementing data preprocessing and reusable to load several datasets, and (b) the configuration file specifying arguments for instantiating a particular dataset variant. Adding variants of other modules (e.g. unlearning method trainers, benchmarks, evaluation metrics etc.) involves a similar procedure.

OpenUnlearning provides a modular framework where most of the Python implementation for dataset classes is shared across various dataset configurations and benchmarks. It also allows users to define custom dataset classes following the steps presented in Figure 5. We already support three commonly used dataset handlers, each serving a distinct purpose in the unlearning pipeline:

- PretrainingDataset: used for training models on large-scale web corpora; essential for simulating pre-training settings.
- CompletionDataset: used for evaluating model outputs in a zero-shot or few-shot setting. This format is particularly useful for measuring memorization and information leakage, such as verbatim reproduction of forgotten content.
- QADataset: designed for probing models using natural language question-answer interactions, optionally with few-shot examples. This format is critical for assessing whether the model retains or forgets factual knowledge in interactive settings. Moreover, the framework automatically pipelines model-specific input formatting such as including system prompts or special tokens for chat-based models ensuring that queries are executed in a manner consistent with the model's native interface.
- ForgetRetainDataset: The unlearning process involves simultaneous optimization on both the forget and retain datasets, requiring concurrent batch loading. This dataset class abstracts this by loading the retain dataset in the same order as the forget dataset for unlearning.

---

[3]We use the same gpt-4-1106-preview endpoint and prompts for data generation.

## C.3 Metrics

`OpenUnlearning` supports multiple evaluation metrics and shares common functionalities across metric implementations. Metrics are broadly classified into three categories and summarized below:

**Memorization Metrics:** These metrics quantify how much the model has memorized information from its training data.

1. **Exact Memorization (EM):** Quantifies memorization by calculating proportion of tokens in the model's response that exactly match those in the ground truth $y$ [58]. Formally, it is defined as

$$\text{EM} = \frac{1}{|y|} \sum_k \mathbf{1} \left\{ \arg\max_y f(y \mid [x, y^{<k}]; \boldsymbol{\theta}) = y^k \right\}, \tag{5}$$

2. **Extraction Strength (ES):** Quantifies the intensity of memorization by determining the minimal prefix length required to reconstruct the remaining suffix [5].

$$\text{ES} = 1 - \frac{1}{|y|} \min_k \left\{ k \mid f([x, y^{<k}]; \boldsymbol{\theta}) = y^{>k} \right\}. \tag{6}$$

3. **Probability (Prob.):** Directly quantifies the model's confidence in its output.

$$\text{Probability} = p\big(f(y \mid x)\big) \tag{7}$$

4. **Paraphrased Probability (Prob.):** Probability computed on a paraphrased answer $y^{\text{para}}$ to remove template bias.

$$\text{Para. Prob.} = p\big(f(y^{\text{para}} \mid x)\big) \tag{8}$$

5. **ROUGE/Paraphrased ROUGE:** Assesses the degree of overlap between the model's output $f(x)$ and the ground truth $y$ [35]. This can be computed against many variants of datasets, including paraphrases and jailbreak prompts (next).

6. **Jailbreak ROUGE:** To probe for forgotten information, we employ a prefix-based jailbreaking attack by prompting the model with *"Sure, here is the answer:"* (as in [63]) and then computing the ROUGE score between the model's response and the ground truth. This metric captures the extent to which suppressed content can still be recovered through prompt manipulation.

7. **Truth Ratio:** Measures the model's preference for the correct answer over a perturbed (incorrect) alternative by comparing their predicted probabilities. A higher value indicates stronger confidence in the correct response. It is defined as:

$$\text{Truth Ratio} = \frac{p(y^{\text{para}} \mid x)}{p(y^{\text{para}} \mid x) + p(y^{\text{pert}} \mid x)} \tag{9}$$

where $y^{\text{para}}$ denotes the paraphrased correct answer and $y^{\text{pert}}$ represents an incorrect alternative with similar structure. Note that Maini et al. [39] use a privacy-oriented variant of Truth Ratio computed as $\text{Truth Ratio} = min(\frac{p(y^{\text{para}}|x)}{p(y^{\text{pert}}|x)}, \frac{p(y^{\text{pert}}|x)}{p(y^{\text{para}}|x)})$. We modify it so that it quantifies extent of knowledge for our work's purposes.

**Privacy Metrics:** These metrics ascertain whether sensitive information from the forget set can still be inferred or extracted from the model. Techniques such as Membership Inference Attacks (MIA) are utilized to evaluate the model's susceptibility to revealing whether specific data points were part of its training set, thereby assessing the privacy guarantees post-unlearning. However, these metrics often assume access to perfectly i.i.d. holdout splits or to an "oracle" retain model, limiting their practical usefulness in real-world settings.

1. **MIA:** Evaluates a model's tendency to memorize training data by testing whether an adversary can distinguish between seen examples from the forget set ($\mathcal{D}_{\text{forget}}$) and unseen examples from a holdout set ($\mathcal{D}_{\text{holdout}}$), based on model confidence. Ideally, a model that has not seen the forget set should yield an AUC of 0.5; however, due to challenges in constructing perfect holdout splits, benchmarks such as MUSE often calibrate this with AUC scores from the retain model (e.g., as done in PrivLeak). We support several MIA methods, including: LOSS [74], ZLib [5], GradNorm [62], MinK [51], and MinK++ [75].

2. **Forget Quality:** Performs a statistical test on the truth ratio distributions of the unlearned and retain models, yielding high values when the distributions closely match.

$$\text{KS}(\text{Truth Ratio}(f_{\text{target}}, \mathcal{D}_f), \text{Truth Ratio}(f_{\text{retain}}, \mathcal{D}_f)) \tag{10}$$

**Utility Metrics:** The goal of unlearning is to effectively forget the targeted data while preserving the model's performance on non-forget data. Utility metrics assess whether the model retains its capabilities on broader tasks beyond the retain data, ensuring that unlearning does not degrade general performance on real-world distributions.

1. **Model Utility (MU):** Captures the retained performance of a model after unlearning, both on the closely tied retain set and on broader general knowledge. TOFU computes MU as the harmonic mean of nine metrics across three data levels: the retain set, real authors, and factual world knowledge. At each level, it evaluates three metrics—probability, ROUGE, and the Truth Ratio. item **ROUGE for knowledge:** MUSE and TOFU assess utility by measuring ROUGE on knowledge-based questions.

2. **Forget Fluency:** Prior work [40, 18] has shown that unlearning often degrades model fluency, particularly on the forget set, resulting in random or nonsensical outputs. To capture this effect, we employ a classifier-based score that predicts whether a given text resembles gibberish[4].

3. **LM Eval Harness**: LM Evaluation Harness [21] is an easy to use library enabling evaluations for a wide variety of general LLM benchmarks. It is integrated into `OpenUnlearning`, unlocking a broad suite of metrics such as WMDP MCQ, MMLU [26], GSM8K [8] etc., for comprehensive post-unlearning evaluation.

By integrating the diverse metrics listed in Table 1, `OpenUnlearning` offers a robust framework to holistically evaluate unlearning methods, ensuring that models not only forget specific data but also maintain utility and privacy standards. Figure 6 illustrates the process of adding a new metric to the `OpenUnlearning` framework.

It is important to recognize that the applicability of unlearning metrics often depends on the dataset used during evaluation. As a result, metrics implemented for one benchmark may not directly transfer to another. For example, the Knowledge Memorization metric in MUSE is based on question-answer pairs where answers are typically short, single-word responses. In contrast, TOFU lacks such a data split and instead features more descriptive, verbose answers. In this context, metrics like ROUGE recall may inadvertently capture surface-level template patterns rather than the core semantic content, potentially misleading the evaluation.

## C.4   Models

Different language models encode and store knowledge in fundamentally different ways depending on their architecture and training setup. As a result, evaluating unlearning methods across a diverse range of models is essential for assessing their robustness and generalizability. However, existing benchmark implementations often support only a narrow set of model types and require users to manually rewrite evaluation logic such as input formatting, tokenization, and prompting—when adapting to new architectures. For example, chat-based models rely on specialized prompting structures that differ significantly from standard causal language models, making adaptation tedious and error-prone.

`OpenUnlearning` supports multiple model architectures and sizes out of the box. Built on Hugging Face Transformers [68], it uses `AutoModelForCausalLM` and `AutoTokenizer`, while also supporting custom model loading (e.g., for probe models). A unified abstraction allows seamless switching between chat-style and base models without modifying the unlearning or evaluation pipeline, reducing overhead and enabling consistent cross-model comparisons.

In addition to support loading models in multiple precisions, `OpenUnlearning` also support loading 4-bit and 8-bit quantized models using the `bitsandbytes` library Dettmers et al. [9]. This flexibility for quantization is particularly valuable for stress testing unlearning Zhang et al. [77].

**New models for TOFU** : `OpenUnlearning` provides trained models for the TOFU benchmark using LLAMA-based architectures finetuned on the TOFU dataset. These models span a range of sizes including 1B, 3B, and 8B parameters, enabling users to explore unlearning behavior across different model capacities. The 1B model, in particular, offers a highly efficient option for rapid experimentation with turnaround time of 15 minutes, requiring only 20 GB of GPU VRAM.

---

[4]`https://huggingface.co/madhurjindal/autonlp-Gibberish-Detector-492513457`

**(a) Metric Handler**

```python
@unlearning_metric(name="rouge")
def rouge(model, **kwargs):
    tokenizer = kwargs["tokenizer"]
    data = kwargs["data"]
    collator = kwargs["collators"]
    batch_size = kwargs["batch_size"]
    generation_args = kwargs["generation_args"]
    ... # calculate ROUGE
    return {
        "agg_value": np.mean(rouges),
        "value_by_index": rouges,
    }
```

**(b) Metric Configuration**

```yaml
# @package eval.muse.metrics.forget_verbmem_ROUGE
defaults: # fill up forget_verbmem_ROUGE's inputs' configs
  - ../../data/datasets@datasets: MUSE_forget_verbmem
  - ../../collator@collators: DataCollatorForSupervisedDatasetwithIndex
  - ../../generation@generation_args: default
handler: rouge # the handler we defined above in (a)
rouge_type: rougeL_f1
batch_size: 8
datasets:
  MUSE_forget_verbmem:
    args:
      hf_args:
        path: muse-bench/MUSE-Books
      predict_with_generate: True
collators:
  DataCollatorForSupervisedDataset:
    args:
      padding_side: left # for generation
generation_args:
  max_new_tokens: 128
```

Figure 6: Example of a metric definition in `OpenUnlearning`: (a) the Python handler that implements the ROUGE metric, and (b) the corresponding configuration used to run ROUGE-based evaluation for assessing verbatim memorization.

Table 4: Supported LLM Architectures in `OpenUnlearning`

| Model | Reference |
|---|---|
| LLAMA-2 | Touvron et al. [59] |
| LLAMA-3.1 / 3.2 | Grattafiori et al. [23] |
| PHI-1.5 | Li et al. [34] |
| PHI-3.5 | Abdin et al. [1] |
| GEMMA | Gemma Team et al. [22] |
| ZEPHYR | Tunstall et al. [60] |
| QWEN-2.5 | Qwen et al. [45] |

## C.5 Unlearning Methods

Unlearning methods form the core of the `OpenUnlearning` framework. In practice, researchers proposing new unlearning approaches often evaluate them on a single benchmark due to the high efforts of adapting their code to other frameworks. This fragmentation has led to a lack of comprehensive, cross-benchmark comparisons in the unlearning literature. The overhead of re-implementing methods, adapting to different evaluation pipelines, and aligning metrics discourages reproducibility and slows progress.

**(a) LLAMA 3.2 1B model configuration**

```
model_args:
  pretrained_model_name_or_path: "meta-llama/Llama-3.2-1B-Instruct"
  attn_implementation: 'flash_attention_2'
  torch_dtype: bfloat16
tokenizer_args:
  pretrained_model_name_or_path: "meta-llama/Llama-3.2-1B-Instruct"
template_args:
  apply_chat_template: True
  system_prompt: You are a helpful assistant.
  date_string: 10 Apr 2025
```

**(b) LLAMA 2-7B model configuration**

```
model_args:
  pretrained_model_name_or_path: "meta-llama/Llama-2-7b-hf"
  attn_implementation: 'flash_attention_2'
  torch_dtype: bfloat16
tokenizer_args:
  pretrained_model_name_or_path: "meta-llama/Llama-2-7b-hf"
template_args:
  apply_chat_template: False
  user_start_tag: "Question: "
  user_end_tag: "\n"
  asst_start_tag: "Answer: "
  asst_end_tag: "\n\n"
```

Figure 7: Example model configurations for two different LLAMA variants: (a) LLAMA 3.2-1B with chat template prompting, and (b) LLAMA 2-7B with manual prompt formatting.

`OpenUnlearning` addresses this gap by providing a unified and modular infrastructure that abstracts away benchmark-specific details. Researchers can implement their method once, typically by extending a custom `Trainer`, and instantly evaluate it across multiple benchmarks. This design dramatically lowers the barrier to method development, evaluation and encourages the community to develop robust methods that work across benchmarks. We currently support all commonly used baselines as well as several state-of-the-art methods, and we invite the community to build upon this foundation.

**Gradient Ascent [39]:** Performs gradient ascent on the forget set to degrade model confidence on targeted data.

$$\mathcal{L} = -\gamma \mathbb{E}_{(x,y_{\mathrm{f}}) \sim \mathcal{D}_{\mathrm{forget}}} \ell\big(y_{\mathrm{f}}|x; f_{\mathrm{unl}}\big) \tag{11}$$

**GradDiff [39]:** Performs gradient ascent on forget data and descent on retain data.

$$\mathcal{L} = -\gamma \mathbb{E}_{(x,y_{\mathrm{f}}) \sim \mathcal{D}_{\mathrm{forget}}} \ell\big(y_{\mathrm{f}}|x; f_{\mathrm{unl}}\big) + \alpha \mathbb{E}_{(x,y) \sim \mathcal{D}_{\mathrm{retain}}} \ell\big(y|x; f_{\mathrm{unl}}\big)$$

**IdkNLL [39]:** Trains to output "I don't know" responses when queried on forgotten content.

$$\mathcal{L} = \gamma \mathbb{E}_{(x,y_{\mathrm{f}}) \sim \mathcal{D}_{\mathrm{forget}}} \ell\big(y_{\mathrm{idk}}|x; f_{\mathrm{unl}}\big) + \alpha \mathbb{E}_{(x,y) \sim \mathcal{D}_{\mathrm{retain}}} \ell\big(y|x; f_{\mathrm{unl}}\big)$$

**IdkDPO [39]:** Uses a DPO-style objective to align the model to output "I don't know" responses when queried on forgotten content.

$$\mathcal{L} = -\frac{2}{\beta} \mathbb{E}_{(x,y_{\mathrm{f}}) \sim \mathcal{D}_{\mathrm{forget}}} \log \sigma\Big( -\beta \log \Big( \frac{p(y_{\mathrm{idk}}|x; f_{\mathrm{unl}})}{p(y_{\mathrm{idk}}|x; f_{\mathrm{target}})} \Big) - \beta \log \Big( \frac{p(y_{\mathrm{f}}|x; f_{\mathrm{unl}})}{p(y_{\mathrm{f}}|x; f_{\mathrm{target}})} \Big) \Big)$$
$$+ \alpha \mathbb{E}_{(x,y) \sim \mathcal{D}_{\mathrm{retain}}} \ell\big(y|x; f_{\mathrm{unl}}\big)$$

**NPO [76]:** Similar to the DPO-style objective, but uses only the negative feedback term in its formulation. It demonstrates better training stability compared to similar methods like GradDiff.

$$\mathcal{L} = -\frac{2}{\beta}\mathbb{E}_{(x,y_{\mathrm{f}})\sim\mathcal{D}_{\mathrm{forget}}}\log\sigma\Big(-\beta\log\Big(\frac{p(y_{\mathrm{f}}|x;f_{\mathrm{unl}})}{p(y_{\mathrm{f}}|x;f_{\mathrm{target}})}\Big)\Big)$$
$$+ \alpha\mathbb{E}_{(x,y)\sim\mathcal{D}_{\mathrm{retain}}}\ell\big(y|x;f_{\mathrm{unl}}\big)$$

**SimNPO [16]:** A modified variant of NPO that retains its core forgetting behavior **by** replacing the reference model with $\delta$ in the loss formulation.

$$\mathcal{L} = -\frac{2}{\beta}\mathbb{E}_{(x,y_{\mathrm{f}})\sim\mathcal{D}_{\mathrm{forget}}}\log\sigma\Big(-\frac{\beta}{|y_{\mathrm{f}}|}\log p(y_{\mathrm{f}}|x;f_{\mathrm{unl}})-\delta\Big)\Big) + \alpha\mathbb{E}_{(x,y)\sim\mathcal{D}_{\mathrm{retain}}}\ell\big(y|x;f_{\mathrm{unl}}\big)$$

**AltPO [40]:** Uses a DPO-style objective to align the model toward generating alternate, in-domain plausible facts (produced by the model itself) that introduce ambiguity and suppress the original target knowledge.

$$\mathcal{L} = -\frac{2}{\beta}\mathbb{E}_{(x,y_{\mathrm{f}})\sim\mathcal{D}_{\mathrm{forget}}}\log\sigma\Big(-\beta\log\Big(\frac{p(y_{\mathrm{alt}}|x;f_{\mathrm{unl}})}{p(y_{\mathrm{alt}}|x;f_{\mathrm{target}})}\Big)-\beta\log\Big(\frac{p(y_{\mathrm{f}}|x;f_{\mathrm{unl}})}{p(y_{\mathrm{f}}|x;f_{\mathrm{target}})}\Big)\Big)$$
$$+ \alpha\mathbb{E}_{(x,y)\sim\mathcal{D}_{\mathrm{retain}}}\ell\big(y|x;f_{\mathrm{unl}}\big)$$

**RMU [33]:** Assumes knowledge is encoded in model parameters and manipulates these representations to suppress memorization signals for the forget set while preserving knowledge in the retain set. Let $\phi(s;f_{\mathrm{unl}})$ denote the embedding features of the model, the loss is given by

$$\mathcal{L} = \mathbb{E}_{(x,y_f)\sim\mathcal{D}_{\mathrm{forget}}}\frac{1}{|y_{\mathrm{f}}|}\sum_{i=1}^{|y_{\mathrm{f}}|}||\phi([x,y^{<i}];f_{\mathrm{unl}})-c\cdot\boldsymbol{u}||_2^2$$
$$+ \mathbb{E}_{(x,y)\sim\mathcal{D}_{\mathrm{retain}}}\frac{1}{|y|}\sum_{i=1}^{|y|}||\phi([x,y^{<i}];f_{\mathrm{unl}})-\phi([x,y^{<i}];f_{\mathrm{target}})||_2^2,$$

where $\boldsymbol{u}$ has elements randomly sampled from $[0,1)$ and $c$ is a scaling hyper-parameter.

**UNDIAL [11]:** Mitigates the instability found in prior methods by employing self-distillation, where the model learns from its own adjusted outputs. The core idea is to reduce the model's confidence in the target token by adjusting its logits, thereby diminishing its influence without affecting the overall model performance. This is achieved by minimizing the KL divergence between the adjusted logits and the model's current output distribution.

$$z_{\mathrm{adj}}(x) = z_{\mathrm{orig}}(x) - \beta\cdot\mathbf{1}_{y_f}$$
$$\mathcal{L} = \gamma\mathbb{E}_{(x,y_f)\sim\mathcal{D}_{\mathrm{forget}}}\Big[\mathrm{KL}\big(\mathrm{softmax}(z_{\mathrm{adj}}(x))\,\|\,\mathrm{softmax}(z_{\mathrm{unl}}(x))\big)\Big] + \alpha\mathbb{E}_{(x,y)\sim\mathcal{D}_{\mathrm{retain}}}\ell\big(y|x;f_{\mathrm{unl}}\big)$$

Where $z_{\mathrm{orig}}(x)$ is the original logits produced by the model before unlearning and $z_{\mathrm{adj}}(x)$ is the adjusted logits.

### C.6 Technical improvements:

**Efficiency:** MUSE evaluates models without batching, while our implementation uses batched inference to improve efficiency. TOFU pads all sequences to a fixed `max_length` of 512, resulting in unnecessary GPU memory and compute overhead. In contrast, we apply dynamic padding based on the longest sequence in each batch. WMDP lacks a rigorous training and unlearning framework, limiting its extensibility for developing and evaluating new methods.

**Training paradigms supported:** Training or unlearning with larger models (e.g., $\geq$ 8B parameters) presents a significant computational challenge, often necessitating multiple high-end GPUs such as NVIDIA A100s. To accelerate this process, we support:

1. **DeepSpeed ZeRO Stage-3** [28]: Enabled via the Accelerate library [25], reducing the memory usage through optimizer state partitioning and CPU/NVMe offloading.
2. **Model Parallelism**: Splits the model across GPUs along its layers, allowing large models to be trained even when individual GPUs cannot hold the full model in memory.

## D  Experimental setup

All subsequent meta-evaluation and benchmarking experiments use the LLAMA-3.2-1B model. Experiments use BF16 precision, a single NVIDIA A100 GPU, a batch size of 32 and a paged AdamW optimizer (matching the TOFU paper's default settings).

## E  Meta-evaluation

### E.1  Faithfulness test-bed design

We create two pools of models: the negative $N$ and the positive $P$ pool. $N$ contains models trained with varying training parameters while avoiding the knowledge of the forget set in the training data. $P$ contains models trained similarly to $N$ but with the target knowledge included in training. During the model pool preparation, we modify the training data used in the $N$ and $P$ pools with several training data variants. This introduces model diversity, forcing metrics to detect genuine *knowledge* retention rather than non-knowledge related artifacts, to achieve high scores. The faithfulness evaluation pipeline is illustrated in Figure 3 (a).

1. **Positive pool ($P$):** Models are trained on all TOFU facts (both `forget10` and `retain90`). We then replace `forget10` with two transformed variants. First, `forget10_paraphrased` uses paraphrased labels while preserving factual content. Second, `forget10_bio` contains long-form biographies derived from `forget10`.

2. **Negative pool ($N$):** Models are trained on the `retain90` split of TOFU, along with two perturbed variants of `forget10`. First, `forget10_perturbed` pairs each forget prompt with an incorrect label. Second, `celeb_bio` (biographies of random celebrities) serves as the counterpart to `forget10_bio`.

To further diversify the model pool, we vary training hyperparameters: five learning rates from $1 \times 10^{-5}$ to $5 \times 10^{-5}$, and two checkpoints (after training epochs 5 and 10). Combining 2 pools $\times$ 3 dataset variants $\times$ 5 learning rates $\times$ 2 checkpoints yields 60 models in total.

**Data generation process**  While some of TOFU's evaluation datasets include paraphrased and perturbed examples, our training-set variants for the model pool were generated independently. We used LLAMA 3.1 405B via the SambaNova API[5] to paraphrase and perturb QA pairs, and prompted Gemini[6] to produce Wikipedia-style biographies from each author's 20 QA pairs.

### E.2  Robustness setup design

We create a large and diverse pool of unlearned models and a separate set of retain models, which serve as gold-standard references having never been trained on the forget set. The unlearned pool is then subjected to stress-test interventions, to provoke recovery (or inducing) of the forgotten knowledge. These pools serve as our test-bed. For every metric being meta-evaluated, values are recorded on both pools before and after each intervention. The change in a metric's distribution before and after intervention on the unlearned models (along with the change in retain models for normalization) is used to characterize robustness. We use three interventions: *relearning*, *quantization* and *probing*.

1. **Relearning Setup:** We finetune the unlearned model on the full `forget10` dataset for one epoch with a learning rate of $2 \times 10^{-5}$.

2. **Quantization Setup:** We apply 4-bit floating-point quantization using BitsAndBytes [9]. Checkpoints unlearned with a learning rate of $1 \times 10^{-5}$ are chosen, as quantization is most effective at lower learning rates [77].

3. **Probing:** We evaluate layer 11 of the LLAMA-3.2-1B model (16 layers total) using the language-model head from the corresponding `retain90`-trained model. This head is trained with a learning rate of $1 \times 10^{-4}$ on `retain90` for ten epochs.

---

[5]`https://cloud.sambanova.ai/playground`
[6]`gemini-2.0-flash-exp` (accessed 26 April 2025)

Table 5: Robustness meta-evaluation with probing (layer 11)

| Metrics | Probe ↑ |
|---|---|
| Exact Mem. | 1.0 |
| Extr. Strength | 1.0 |
| Truth Ratio | 1.0 |
| Prob. | 0.99 |
| ROUGE | 0.99 |
| Jailbreak ROUGE | 0.99 |
| Para. Prob. | 1.0 |
| Para. ROUGE | 0.99 |
| MIA - LOSS | 1.0 |
| MIA - MinK | 1.0 |
| MIA - MinK++ | 0.83 |
| MIA - ZLib | 1.0 |

## E.3 Additional Results

Figure 8 shows the faithfulness of the metrics, while Figure 9 and Figure 10 show their behavior under relearning and quantization stress tests. We found that removing MU filter of retaining at least 80% utility for unlearned models reduces robustness to quantization further (see Figure 11). Despite this, we apply the MU filter to better align with common unlearning reporting practices.

**Probing results:** We compute the metric robustness to probing intervention as follows

$$p = \frac{m_{\text{ret}}^a}{m_{\text{unl}}^a} \quad \text{if} \quad \frac{m_{\text{ret}}^b}{m_{\text{unl}}^b} \geq 1, \quad P = \min(p, 1) \tag{12}$$

Table 5 shows the results of our metric meta-evaluation with probing. Probing, while provided for by `OpenUnlearning`, is not used in the meta-evaluation procedure, as $P$ scores on TOFU achieve 1 for all metrics and thus offer little information.

## E.4 Further considerations

**Why aren't the intervened versions of metrics considered evaluation metrics themselves?** The interventions we use require modification to and access of model weights, which an unlearning auditor might not possess. In the case of relearning and quantization, they also involve computational costs associated with training and calibration. Stress-testing interventions are best suited for final-stage audits before model deployment, rather than for routine use throughout unlearning workflows, as is expected of standard evaluation metrics. Our analysis can inform the design of robust evaluation metrics that function without requiring stress-testing.

**Comparison to Wang et al. [63]'s meta-evaluation:** Our work is related to the recent effort by Wang et al. [63] to compare unlearning evaluation metrics. Their analysis focuses on four metrics: probability, ROUGE, ES, and EM, and evaluates robustness by measuring the linear correlation of metric values before and after applying stress-tests such as jailbreaking, relearning, probing, and token noising. We extend this framework in several key ways.

1. **Broader metric coverage:** We evaluate a broader range of metrics, including six additional ones.
2. **Faithfulness assessment:** We assess faithfulness of metrics in our meta-evaluation as a minimal criterion. This enforces that good metrics must accurately capture the presence or absence of target knowledge, rather than merely resisting change under intervention.
3. **Focused interventions:** We focus specifically on three interventions: relearning, probing, and quantization, excluding jailbreaking and token-noising from the intervention set. We instead treat jailbreaking as an evaluation metric in its own right. Prompt-based attacks like paraphrasing and jailbreak-style prompts are more naturally seen as inexpensive evaluation metrics rather than stress-testing interventions. Additionally, Wang et al. [63] found jailbreaking and token noising (which is also a prompt modification) to be less effective as interventions.
4. **A different calibration criterion:** Our procedure also introduces a calibration criterion grounded in ideal behavior. Rather than expecting linear variation from a metric upon intervention, we

benchmark metric behavior against a gold-standard retain model, for a more principled signal of robustness.

5. **Practical robustness analysis:** Our robustness analysis filters for models with good utility that are substantially unlearned, selected from a diverse and representative set of unlearning algorithms. This leads to a test distribution for metrics that better reflects realistic unlearning scenarios.

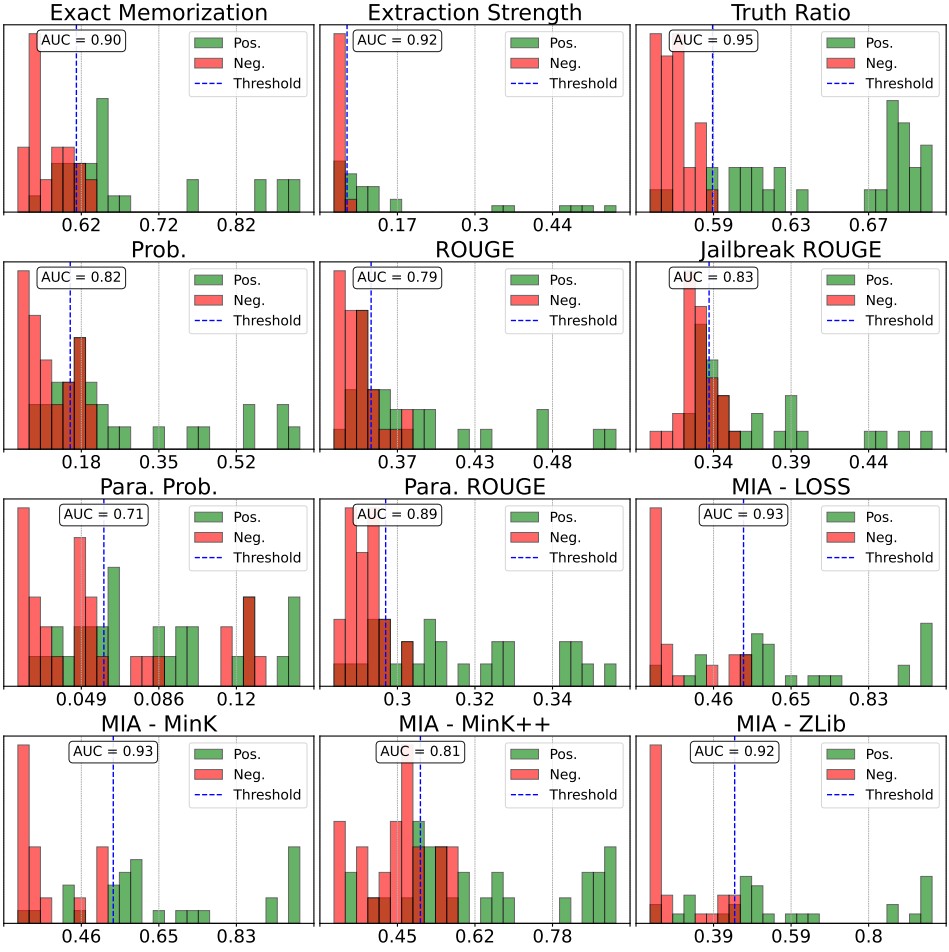

Figure 8: Faithfulness: Evaluation of multiple metrics to assess faithfulness. AUC indicates how effectively metrics distinguish between models trained on the target knowledge and those that are not.

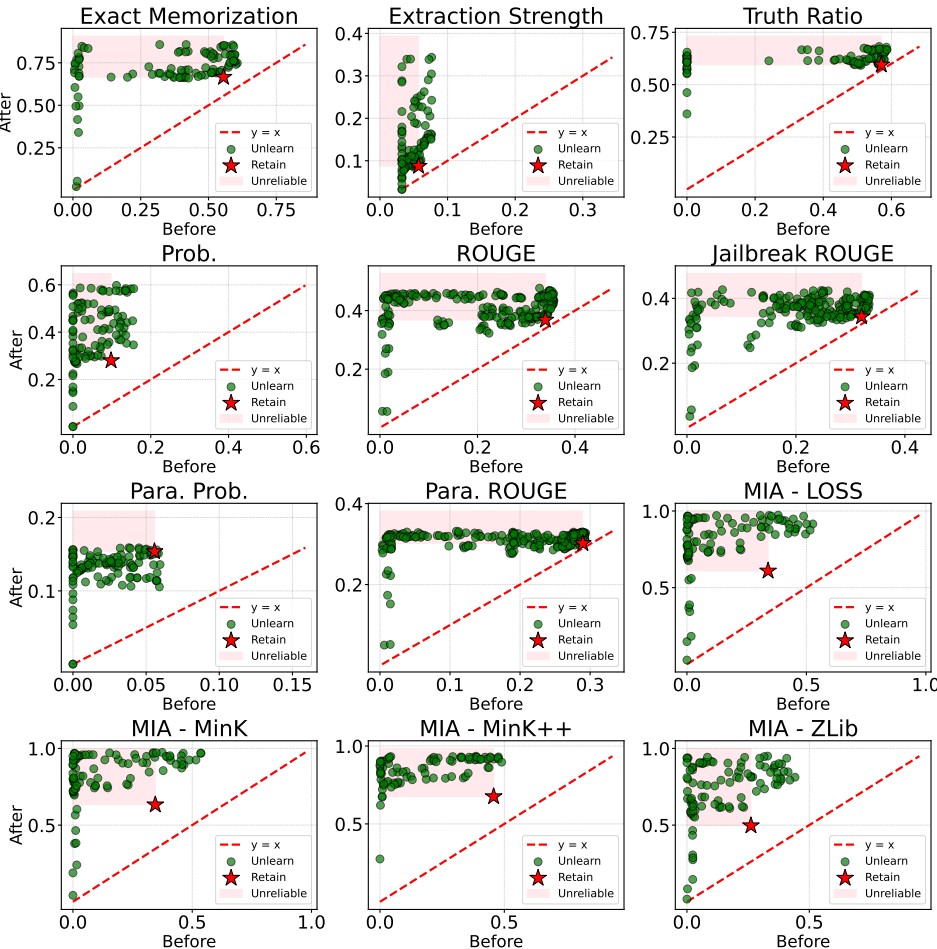

Figure 9: Relearning: Stress-testing multiple evaluation metrics through relearning. A significant fraction of unlearned models regain knowledge faster than the retained model when re-exposed to the forgotten data, falling into the unreliable red-shaded region: indicating that the metrics failed to initially capture the knowledge and are thus not robust.

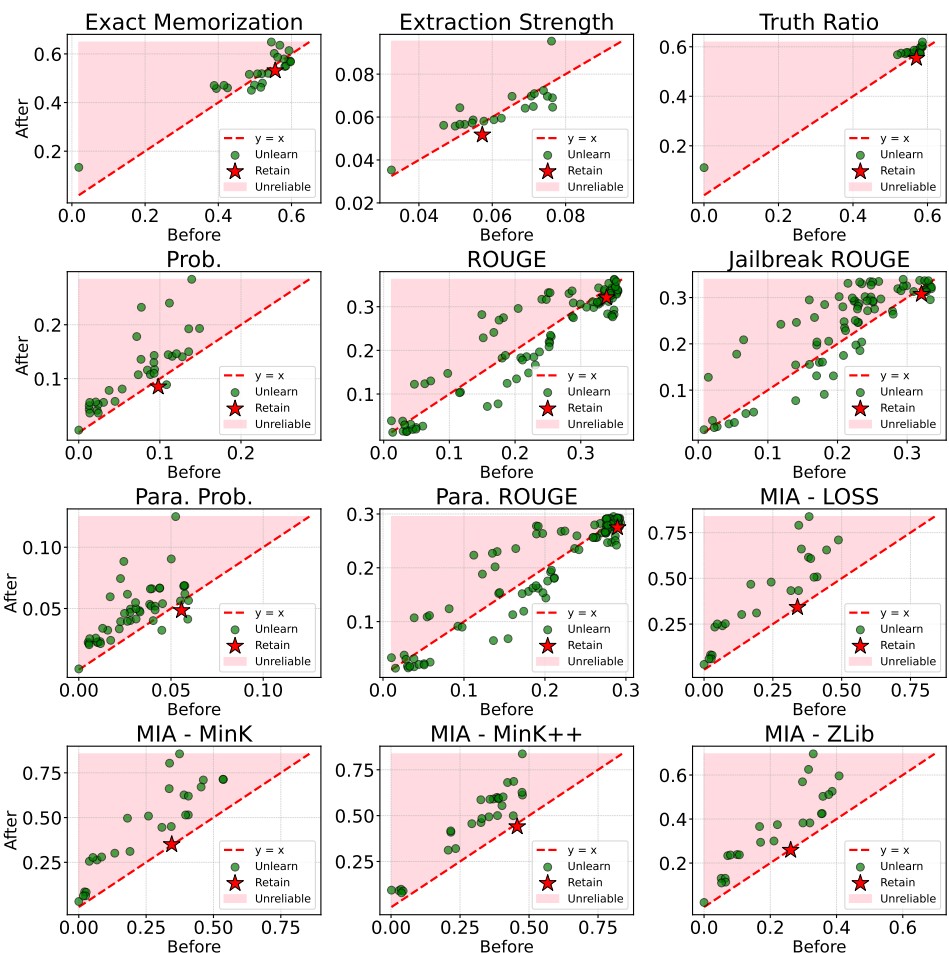

Figure 10: Quantization: Stress-testing multiple evaluation metrics through quantization. For several metrics, a subset of unlearned models shows increased metric values after quantization, falling into the red-shaded region: suggesting that the metrics failed to initially capture the presence of knowledge and are therefore not robust. These results are reported only for models unlearned with low learning rates and high utility.

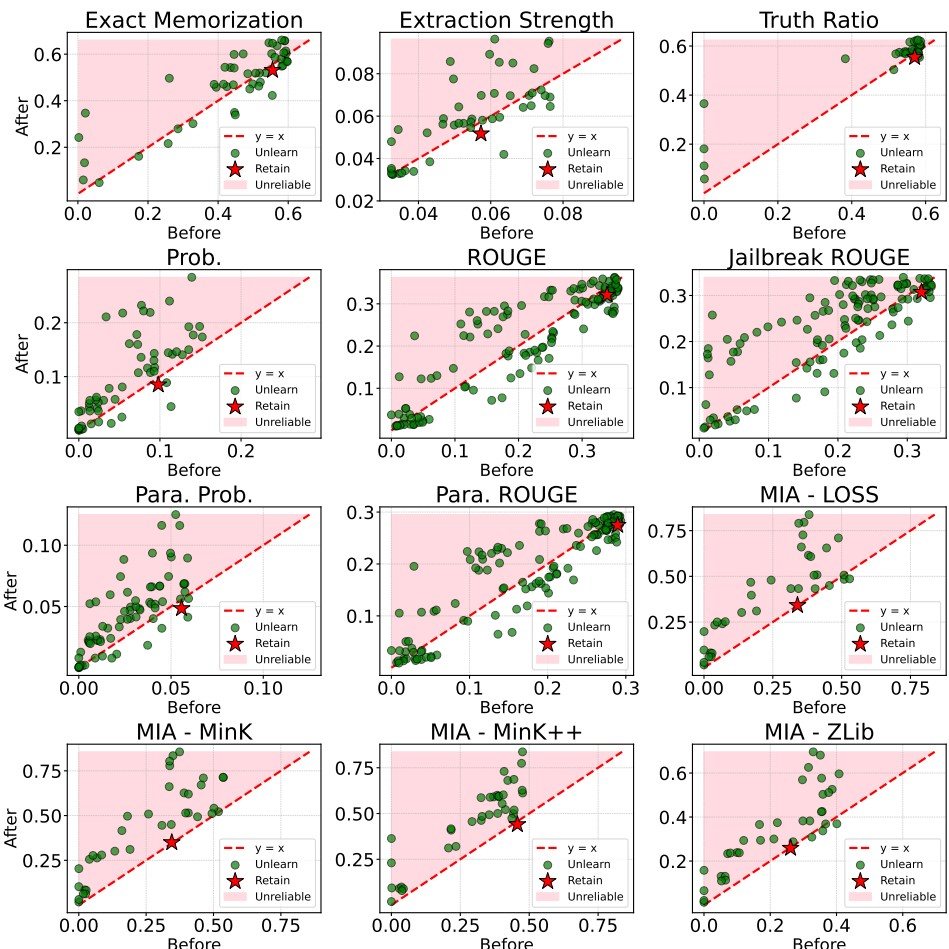

Figure 11: Quantization: Stress-testing multiple evaluation metrics through quantization. For each metric, a subset of unlearned models shows increased metric values after quantization, falling into the red-shaded region, suggesting that the metrics failed to initially capture the presence of knowledge and are therefore not robust. These results are reported only for models unlearned with low learning rates and no filter on utility.

# F Further discussion on benchmarking unlearning methods

## F.1 Metric aggregation

There are theee dimensions evaluated by our suite of metrics 1) Memorization, 2) Privacy 3) Utility.

We consider multiple metrics in each dimension and aggregate the score as follows:

1. **Memorization**: To quantify the degree of successful forgetting, the Memorization Score is calculated as the Harmonic Mean (HM) of 4 core metrics which are best as per our meta-evaluations analysis in §2 — ES, EM, Paraphrased Probability and Truth Ratio. These metrics are inverted (i.e., $1 -$ metric) so that higher scores indicate more effective unlearning. The score is given by:

$$\text{Memorization Score} = \text{HM}\left(1 - \text{ES},\ 1 - \text{EM},\ 1 - \text{Para. Prob},\ 1 - \text{Truth Ratio}\right)$$

2. **Privacy**: For assessing privacy, we utilize four Membership Inference Attack (MIA) metrics: LOSS, ZLib, Min-k, and Mink++. For each of these, an individual privacy score ($s_{\text{MIA}}$) is calculated. This score, ranging from 0 to 1, quantifies how closely the unlearned model's behavior on the specific MIA metric aligns with that of a gold-standard retain model (details below). A higher $s_{\text{MIA}}$ score indicates greater similarity to the retain model. The overall Privacy Score is then the Harmonic Mean (HM) of these individual scores:

$$\text{Privacy Score} = \text{HM}(s_{\text{LOSS}}, s_{\text{ZLib}}, s_{\text{Min-k}}, s_{\text{Mink++}})$$

3. **Utility**: TOFU evaluates a model's utility using nine core metrics that assess performance across splits at three different distances from the forget dataset distribution - namely, retain, real-world authors, and wrong-fact queries: using QA probability, ROUGE, and truth-ratio scores. In addition to this we include a new metric that measures the fluency of the model's response when prompted with entities-related to forget queries, following [40, 18]. Fluency is assessed using a classifier[7] that detects gibberish / nonsensical outputs. The final utility score is the harmonic mean of MU and fluency. Note that we scale all metrics with init finetuned model, so their scores across all points fall in the $[0, 1]$ range. For example, TOFU MU scores never exceed that of the initial target model upon unlearning, so all scores are effectively divided by the target model's MU.

Note that for many metric aggregations we use Harmonic Mean, as HM ensures that a high final score demands strong performance in all constituent parts.

## F.2 Hyperparameter tuning and model selection while comparing unlearning methods

**Hyperparameters used**

1. For GradDiff and IdK-NLL: we vary the learning rate over the set $\{1 \times 10^{-5}, 2 \times 10^{-5}, 3 \times 10^{-5}, 4 \times 10^{-5}, 5 \times 10^{-5}\}$, and sweep the regularization coefficient $\alpha \in \{1, 2, 5, 10\}$.
2. For IdK-DPO, NPO and AltPO: we tune learning rates in $\{1 \times 10^{-5}, 2 \times 10^{-5}, 5 \times 10^{-5}\}$, and search over $\alpha \in \{1, 2, 5\}$ and $\beta \in \{0.05, 0.1, 0.5\}$.
3. For RMU: we use the same learning rate range $\{1 \times 10^{-5}, 2 \times 10^{-5}, 5 \times 10^{-5}\}$, vary the steering coefficient in $\{1, 10, 100\}$, and apply the loss at one of the layers $l \in \{6, 11, 16\}$ of the LLama3.2-1B model. For each selected layer $l$, we restrict training to layers $l - 2$, $l - 1$, and $l$.
4. For SimNPO: we tune learning rates in $\{1 \times 10^{-5}, 2 \times 10^{-5}, 5 \times 10^{-5}\}$, and search over $\beta \in \{3.5, 4.5\}$, $\delta \in \{0, 1\}$ and $\delta \in \{0.125, 0.25\}$.
5. For UNDIAL: we tune learning rates in $\{1 \times 10^{-5}, 1 \times 10^{-4}, 3 \times 10^{-4}\}$, and search over $\alpha \in \{1, 2, 5\}$ and $\beta \in \{3, 10, 30\}$.

We aggregate utility score and memorization score and use their harmonic mean for tuning the models.

**What metrics are appropriate for model selection during hyperparameter tuning?** The nature of tuning in unlearning benchmarking has distinct considerations compared to general machine learning. While standard machine learning avoids using test data for tuning to ensure generalization, unlearning

---

[7]https://huggingface.co/madhurjindal/autonlp-Gibberish-Detector-492513457

Table 6: Comparison of unlearning methods on the TOFU task, showing aggregate (Agg.) using only Memorization (Mem.) and utility (Utility) scores. Privacy scores are not used in the aggregation and are only shown for illustration. Higher scores indicate better performance (↑). Initial finetuned is the target model before unlearning and Retain model is the gold standard target model. The focus on memorization as opposed to privacy results in GradDiff performing the best as it easily results in over-unlearning.

| Method | Agg. ↑ | Mem. ↑ | Priv. ↑ | Utility ↑ |
|---|---|---|---|---|
| Init. finetuned | 0.00 | 0.00 | 0.10 | 1.00 |
| Retain | 0.58 | 0.31 | 1.00 | 0.99 |
| GradDiff [39] | **0.87** | **0.97** | 3.27e-03 | 0.79 |
| AltPO [40] | 0.76 | 0.63 | 0.06 | 0.95 |
| IdkDPO [39] | 0.71 | 0.56 | 0.06 | 0.95 |
| NPO [76] | 0.69 | 0.52 | 0.06 | **0.99** |
| RMU [33] | 0.53 | 0.47 | 0.5 | 0.61 |
| SimNPO [16] | 0.49 | 0.32 | 0.63 | 1.0 |
| UNDIAL [11] | 0.4 | 0.27 | 0.48 | 0.78 |
| IdkNLL [39] | 0.14 | 0.08 | 0.17 | 0.93 |

in TOFU and MUSE specifically targets the known forget set for erasure. Consequently, iteratively refining the unlearning by evaluating the model's behavior concerning this specific set is a permissible approach to ensure thorough forgetting before deployment. For this tuning, we advocate relying on metrics realistically available during the development phase, specifically those assessing forget quality on the target data and general utility, while avoiding "oracle" metrics that presume access to unavailable resources like true i.i.d holdout sets or retain models like in [39, 52]. Since all our privacy scores use a retain model, we avoid them during tuning. We rely on the harmonic mean of the Memorization and Utility scores as the validation objective.

**Comparison to Wang et al. [63]'s benchmarking:** While Wang et al. [63] propose approaches towards model selection and benchmarking through validation on Extraction Strength and calibration via model-merging, their analysis has several limitations. They rely only on ES scores for evaluating forgetting and utility. ES was found to be robust among the set of 4 evaluation metrics (an observation also re-verified in our work (§4). Yet it has not been proved that ES is a comprehensive metric validating all facets of knowledge unlearning. For example, ES does not account for privacy metrics that prevent over-unlearning, like TOFU's Truth Ratio or FQ or MUSE's PrivLeak. In addition, they do not consider all facets of general utility evaluation, particularly forget set fluency. Finally, the question of what metrics can be used in model selection and if they must be separate from the leaderboard metrics remains unanswered. These limitations remain, to a smaller degree, in our benchmarking procedure, and we consider this an important line for further research.

