# OpenReview forum: "OpenUnlearning: Accelerating LLM Unlearning via Unified Benchmarking of Methods and Metrics"
_NeurIPS.cc/2025/Datasets_and_Benchmarks_Track — NeurIPS 2025 Datasets and Benchmarks Track poster_

### Official Review · Reviewer_vWbS · 2025-06-19

**Rating:** 4
**Confidence:** 3

**Summary:**

The paper introduces OpenUnlearning, a unified, open-source framework that standardizes the study of large-language-model (LLM) unlearning. It bundles nine state-of-the-art unlearning algorithms, sixteen evaluation metrics and three widely used benchmarks (TOFU, MUSE and WMDP) into a single, modular pipeline. Beyond consolidating existing tools, the authors propose a meta-evaluation suite that tests whether unlearning metrics themselves are faithful (able to distinguish models that still contain forgotten knowledge) and robust (stable under quantization or relearning). Using this platform they benchmark eight unlearning methods, identify SimNPO and RMU as current front-runners, and release 450 + model checkpoints to spur future research.

**Dataset Code Accessibility:**

No

**Ethical Considerations:**

No, there are no or only very minor ethics concerns

**Limitations Weaknesses:**

Despite its scope, the framework is still bound to the current “forget–retain” paradigm and the same popular benchmarks, which several studies critique as weak or unrealistic proxies for real-world deletion demands; thus OpenUnlearning may not yet capture all unlearning challenges. The authors acknowledge that their faithfulness-and-robustness criteria are only a minimal set—other desiderata such as fairness or computational cost remain unexplored. Finally, their comparative study shows ranking sensitivity to metric selection and hyper-parameter tuning, suggesting that even with a unified platform, unequivocal evaluation of unlearning quality is still an open problem.

**Strengths Contributions:**

OpenUnlearning effectively tackles the field’s fragmentation by offering a “one-stop” infrastructure where new algorithms or metrics can be plugged in once and evaluated everywhere, sharply reducing duplication of effort . Its meta-evaluation of metrics is novel, providing the first systematic evidence that Extraction Strength and Exact Memorization are currently the most reliable forgetting scores, while highlighting weaknesses in popular MIA-based measures
. The framework’s breadth—covering diverse datasets, stress tests (relearning, quantization, probing) and multiple model families—and its public release with rich documentation and pre-configured YAML templates make it an immediately practical resource for both academic and industrial practitioners, fostering reproducible and community-driven progress.

---

> ### Author Rebuttal · Authors · 2025-07-31
>
> We thank you for their comprehensive review and for recognizing that OpenUnlearning "effectively tackles the field's fragmentation" by providing a "one-stop infrastructure." We appreciate your assessment that our meta-evaluation provides "the first systematic evidence" for metric reliability and that our framework represents an "immediately practical resource."
>
> ---
>
> ## *Summary of responses*
>
> | Concern                            | Response                                                      |
> |-----------------------------------|---------------------------------------------------------------|
> | Forget-retain paradigm limitations | OpenUnlearning supports WMDP which is akin to "real-world" unlearning challenges. Additionally, it is designed to seamlessly extend to future paradigms.
> | Limited desiderata criteria        | Faithfulness and robustness are foundational criteria and we enable further expansion.
> | Metric ranking sensitivity         | Our observation actually highlights the need for research into principled evaluation design, strengthening position |
>
> ---
>
> ## 1. Response to Forget-Retain Paradigm Limitations
>
> **OpenUnlearning's positioning:** We would like to emphasize that the WMDP benchmark operates under a relatively realistic set of unlearning goals, where certain "harmful knowledge" must be forgotten from the model, without a single attribution to the source (the forget set here is not "exclusive" to that information).  At the same time, we acknowledge that the forget-retain paradigm faces limitations and most major benchmarks (TOFU, MUSE) operate within this paradigm. Standardizing evaluations within the current landscape is an important first step as we evolve to new paradigms. We envision OpenUnlearning as a living framework that evolves with the field, providing consistent infrastructure as new benchmarks emerge while maintaining backward compatibility with current research.
>
>
> One exciting trend that speaks about the living nature of OpenUnlearning is the recent community contributions since the release. This includes unlearning methods such as RMU, UNDIAL, PDU, CE-U, WGA, SatImp and AltPO, with further new benchmark implementations underway.
>
> ---
>
> ## 2. Response to Limited Desiderata Criteria
>
> You correctly note that there can be further dimensions of meta-evaluation, such as fairness and computational cost, which can be explored. We would like to point out that the faithfulness and robustness criteria we evaluate are very important foundational criteria that any unlearning metric should satisfy. Each of them individually represent necessary conditions for metric quality, whereas optimizing for alternative criteria like computational cost (lowest cost) may not always be ideal. Ultimately, we are dealing with a challenging circular problem of evaluating evaluations that evaluate unlearning methods themselves. We chose this set of criteria as a minimal set of properties that the evaluation metric must satisfy.
>
> ---
>
> ## 3. Response to Ranking Sensitivity Concerns
>
> >their comparative study shows ranking sensitivity to metric selection and hyper-parameter tuning, suggesting that even with a unified platform, unequivocal evaluation of unlearning quality is still an open problem.
>
> Indeed, our results show concrete evidence for substantial ranking sensitivity to design choices. SimNPO and RMU emerge as top performers under our aggregation scheme in our privacy-inclusive ranking (Table 3) vs. memorization-focused ranking (Table 6, Appendix F.2) shows substantial differences
>
> **This is a valuable finding**: The observed sensitivity actually strengthens our core argument for standardized evaluation. By exposing this sensitivity, we demonstrate the bias introduced by arbitrary decisions in leaderboard design. We hope this spurs research in the community towards principled benchmarking and leaderboard design.
>
>
> We hope this addresses your concerns about scope limitations while demonstrating how our work provides immediate value and lays groundwork for future evolution. **Please let us know if any additional analysis in particular would help strengthen your assessment of our contributions.**

---

> > ### Comment · Reviewer_vWbS · 2025-08-04
> >
> > The authors provide thoughtful rebuttals，I will maintain my score.

---

> > > ### Author Response · Authors · 2025-08-05
> > >
> > > Dear Reviewer,
> > >
> > > Thank you for engaging with our rebuttal. In particular, we appreciated the opportunity to clarify that:
> > > - Our framework goes beyond the standard forget-retain paradigm (e.g., through WMDP),
> > > - We proposed two foundational principles that every unlearning metric must uphold, and
> > > - Our ranking sensitivity analysis highlights the fragility of current evaluation practices and drives more community engagement towards this problem.
> > >
> > > If any remaining concerns are holding back a stronger support (beyond borderline accept), we would be grateful to hear them. We genuinely appreciate your thoughtful feedback throughout this process.

---

### Official Review · Reviewer_HtBA · 2025-07-02

**Rating:** 5
**Confidence:** 4

**Summary:**

This paper introduces OpenUnlearning, a unified and extensible framework for benchmarking LLM unlearning methods and evaluation metrics. The framework integrates various state-of-the-art unlearning algorithms and evaluation techniques across multiple benchmarks, enabling comprehensive comparative analysis and fostering reproducible research in the field. The authors also propose a novel meta-evaluation benchmark to assess the faithfulness and robustness of unlearning metrics themselves, contributing significantly to the rigorous development of LLM unlearning research.

**Dataset Code Accessibility:**

Yes

**Ethical Considerations:**

No, there are no or only very minor ethics concerns

**Final Justification:**

The authors' rebuttal has partially addressed my concerns, and the paper is solid and promising to support the unlearning community of LLM.

**Limitations Weaknesses:**

- The paper should include Qwen series models in the supported LLM architectures, which could be a valuable addition for broader model diversity since Qwen models are trending and popular for the LLM community.

- The author should better include the RWKU dataset as the primary benchmarking, as the dataset offers a more realistic approach to unlearning pre-trained knowledge directly from off-the-shelf LLMs, unlike benchmarks that inject new knowledge via finetuning. I understand the authors have already implemented WMDP but RWKU is also neccary for the general use of unlearning pretrained knowledge.

- While the paper discusses the practicality of tuning on privacy metrics, a more in-depth exploration of the trade-offs and potential biases introduced by the chosen tuning strategy for model selection would be beneficial.

- The current meta-evaluation setup, while valuable, could be further expanded to incorporate a wider range of unlearning scenarios, model architectures, and novel methods to comprehensively assess unlearning metrics as the field progresses.

**Strengths Contributions:**

- The introduction of OpenUnlearning is highly timely and well-motivated for the LLM unlearning community, addressing the critical need for a standardized framework to unify fragmented research efforts and accelerate progress in this rapidly evolving field.

- The framework is comprehensive, integrating 9 state-of-the-art unlearning algorithms, 16 diverse evaluations, and 3 leading benchmarks (TOFU, MUSE, and WMDP), facilitating extensive analysis of forgetting behaviors across over 450 publicly released checkpoints.

- The paper's meta-evaluation benchmark for unlearning metrics, focusing on faithfulness and robustness, is a novel and crucial contribution, addressing the inherent challenges in reliably measuring whether unlearning has truly occurred.

---

> ### Author Rebuttal · Authors · 2025-07-31
>
> We appreciate your thoughtful review and assessment that OpenUnlearning is "highly timely and well-motivated." Thank you for recognizing our meta-evaluation benchmark as a "novel and crucial contribution" and for the constructive suggestions to strengthen our framework.
>
> ---
>
> ## *Summary of responses*
>
> | Concern                                | Response                                                                                                                         |
> |----------------------------------------|----------------------------------------------------------------------------------------------------------------------------------|
> | Further benchmark and model support    | Modular design enables straightforward integration of new models and benchmarks. Community contributions power the framework.   |
> | Metric tuning concerns                 | Clarified privacy–memorization trade-offs; will expand discussion in the revision.                                                |
> | Meta-evaluation scope limitations      | Demonstrated extensiveness of current evaluation and outlined roadmap for future extensions.                                      |
>
> ---
>
> ## 1. Response to Qwen Model Architecture Support
>
> You correctly note that the paper should include the Qwen series models given their popularity. This is an excellent suggestion that aligns with our goal of broad architecture coverage.
>
> **Technical feasibility**: Our framework's modular design makes it straightforward to extend to new architectures. We built on HuggingFace Transformers' AutoModel functionality specifically to enable easy extension to new model families.
>
> **Implementation plan**: We have now added a new PR to OpenUnlearning to support Qwen models across our model sizes (1B, 3B, 8B parameter variants similar to our current Llama coverage). We will update Table 4 in the paper to reflect 7 supported architectures rather than 6.
>
> Given Qwen's popularity, this addition will significantly broaden our framework's applicability and adoption within the community.
>
> ---
>
> ## 2. Response to RWKU Benchmark Integration
>
> You raise a valid point that the RWKU dataset offers a more realistic setting for evaluating unlearning directly from off-the-shelf LLMs, and thus could serve as a valuable benchmark.
>
> While we acknowledge the strengths of RWKU, our **initial selection** of TOFU, MUSE, and WMDP was guided by the following considerations:
>
> 1. **Community adoption:** These benchmarks have been widely used in the literature, with multiple papers reporting results, enabling better comparability across methods.
> 2. **Iteration efficiency:** They offer a faster turnaround in terms of both time and computational cost, allowing researchers to iterate more rapidly.
> 3. **Diversity of scenarios:** Our selection spans both real-world (WMDP) and synthetic (TOFU, MUSE) settings to capture a broad range of unlearning challenges.
>
> **Community efforts** We've received community interest to help add additional benchmarks such as KnowUnDo [1], RESTOR [2] etc. Our framework is designed for extensibility in making new contributions. We wish to facilitate benchmark owners to smoothly integrate their work in OpenUnlearning, through technical support and our extensive step-by-step documentation.
>
> **Implementation plan for RWKU:** We have already reached out to the original authors behind RWKU about their permission and/or interest in adding the benchmark to OpenUnlearning. We are actively working out the plan!
>
> [1]. Tian, Bozhong, et al. "To forget or not? towards practical knowledge unlearning for large language models." arXiv preprint arXiv:2407.01920 (2024).
> [2] Rezaei, Keivan, et al. "RESTOR: Knowledge Recovery in Machine Unlearning." arXiv preprint arXiv:2411.00204 (2024).
>
> ---
>
> ## 3. Response to Metric Tuning Concerns
>
>
> > a more in-depth exploration of the trade-offs and potential biases introduced by the chosen tuning strategy for model selection.
>
> We agree this is important and clarify our reasoning.
>
> **Privacy vs Memorization Tradeoff**: Memorization metrics penalize high likelihood on forget data; privacy metrics penalize both unusually high *and* low likelihoods. Thus, strong unlearning (e.g., GradDiff with memorization score 0.97) can *hurt* privacy (score 3e-3) due to over-unlearning—assigning lower likelihood than even a retain model.
>
> **Why we didn’t tune on privacy**:
>
> * Privacy metrics assume access to a retain model and IID forget-like holdout data. This is unrealistic in many cases (such as WMDP, and othe real world scenarios).
> * Tuning on memorization alone risks over-unlearning.
> * We chose to tune on memorization + utility and explain this in Appendix F.2.
>
> Practitioners can still apply privacy-based tuning by approximating our metric setup (e.g., using a model trained without forget data as a retain proxy and creating a synthetic holdout).
>
> We will expand this clarification in the revision. We will in particular add a discussion on this tradeoff, as verified by concrete numbers in our evaluation.
>
> ---
>
> ## 4. Response to Meta-Evaluation Scope Limitations
>
> >current meta-evaluation setup, while valuable, could be further expanded to incorporate a wider range of unlearning scenarios, model architectures, and novel methods.
>
> **Extensive nature of current experimentation**: Our current experimentation incurred a substantial computational cost: we trained and released 450 models, while meta-evaluating 12 metrics across 3 stress testing setups. Therefore, we focused on a single scenario (TOFU) and architecture (Llama-3.2-1B) due to their significant advantages in scalability and computational efficiency, which enabled the release of 450+ models.
>
>
> **Future roadmap**: At the same time, we are committed to broadening our scope and believe the community can use our framework to easily extend this further. Our framework enables easy addition of newer unlearning methods, architectures and benchmarks, as demonstrated by our progress so far. One exciting trend that speaks about the living nature of OpenUnlearning is the recent community contributions since the release. This includes unlearning methods such as RMU, UNDIAL, PDU, CE-U, WGA, SatImp and AltPO, with further new benchmark implementations underway.
>
>
>
> ---
>
> We hope this addresses your suggestions and demonstrates our commitment to making OpenUnlearning a continuously improving resource that evolves with the field's needs. **Please let us know if any additional analysis in particular would help strengthen your assessment of our contributions.**

---

> > ### Comment · Reviewer_HtBA · 2025-08-04
> >
> > The authors' rebuttal has partially addressed my concerns, and the paper is solid and promising to support the unlearning community of LLM. Therefore, I keep my score.

---

### Official Review · Reviewer_QUKS · 2025-07-03

**Rating:** 5
**Confidence:** 3

**Summary:**

The paper proposes OpenUnlearning, a unified, standardized, and extensible framework for benchmarking LLM unlearning methods and evaluations. The proposed framework integrates nine unlearning algorithm and 16 evaluations, and analyze over 450 publicly released checkpoints. The proposed benchmark mainly focus on  assessing the faithfulness and robustness of evaluations. The project is released on GitHub and have a substantially impact.

**Dataset Code Accessibility:**

Yes

**Ethical Considerations:**

No, there are no or only very minor ethics concerns

**Final Justification:**

All my concerns have been addressed. I would like to maintain this score as it is very positive. Thank you for the hard work of the authors.

**Limitations Weaknesses:**

1. I have a question on the data side of unlearning, which is not discussed in the paper. LLMs are usually trained on massive datasets (over trillions of tokens). How can we identify the specific collection of data samples that need to be forgotten? I know there are existing works on influence estimation [1, 2, 3, 4] and data valuation [1, 5], but I am not sure if they are directly related.
2. As shown in Figure 3, the framework seems to assume that the forget set is already known in the problem setting. However, identifying the forget set can be very difficult. The paper also mentions, "The setup usually also involves a retain set disjoint from the forget set" (Line 67). I believe that determining these disjoint sets is quite challenging and should be discussed in the paper.


[1] Choe, Sang Keun, et al. "What is your data worth to gpt? llm-scale data valuation with influence functions." arXiv preprint arXiv:2405.13954 (2024).

[2] Lin, Huawei, et al. "Token-wise Influential Training Data Retrieval for Large Language Models." arXiv preprint arXiv:2405.11724 (2024).

[3] Li, Zhe, et al. "Do Influence Functions Work on Large Language Models?." arXiv preprint arXiv:2409.19998 (2024).

[4] Grosse, Roger, et al. "Studying large language model generalization with influence functions." arXiv preprint arXiv:2308.03296 (2023).

[5] Pan, Yanzhou, et al. "ALinFiK: Learning to Approximate Linearized Future Influence Kernel for Scalable Third-Party LLM Data Valuation." arXiv preprint arXiv:2503.01052 (2025).

**Strengths Contributions:**

1. The proposed OpenUnlearning framework is unified, standardized, and extensible, making it beneficial for advancing research in this area.
2. The framework evaluates both unlearning methods and the metrics themselves, with a particular focus on their faithfulness and robustness.
3. The paper presents extensive experiments conducted across three unlearning benchmarks: TOFU, MUSE, and WMDP.
4. The project has demonstrated significant impact, as evidenced by its strong presence and engagement on GitHub.
5. I enjoyed reading the paper, and it truly inspired me.

---

> ### Author Rebuttal · Authors · 2025-07-31
>
> Thank you for your thoughtful review and for recognizing OpenUnlearning as "unified, standardized, and extensible." We're delighted that our work "truly inspired" you and that you see the value in evaluating both unlearning methods and the metrics themselves.
>
> ---
>
> ## *Summary of responses*
>
> | Concern                          | Response                                                                                                                          |
> |----------------------------------|-----------------------------------------------------------------------------------------------------------------------------------|
> | Relation to influence estimation | Complements influence‐estimation research by focusing on *how* to forget and evaluate forgetting.                                  |
> | Dataset identification challenges| Outlined guidelines for identifying forget and retain sets. |
> | Draft positioning                | Will update draft to detail the end‐to‐end pipeline and position our work within the broader unlearning ecosystem.                |
>
> ---
>
>
> ## Response to Dataset Identification Challenges
>
> Your observation about the gap between our evaluation framework and practical data identification challenges is well-taken. We view this as highlighting the need for end-to-end unlearning pipelines that combine different stages:
>
> 1. **Identification**: Using influence estimation methods to identify what to forget
> 2. **Unlearning**: Using unlearning methods implemented and benchmarked in our framework to forget the data
> 3. **Validation**: Using stress testing and evaluation metrics provided by our framework and validated in our meta-evaluation to ensure forgetting is effective and robust
>
> ### **Complementary approaches**
> Methods like influence functions (Choe et al. [1], Grosse et al. [4]) and data valuation (Pan et al.) address *what* to forget (step 1), while our framework addresses *how* to forget effectively and *how* to evaluate that forgetting once targets are identified  (steps 2-3). The influence estimation works you cite are highly relevant for the upstream identification problem.
>
> ### **Determining splits**
> - **Forget set**: Practitioners would use forget sets identified in requests (e.g., GDPR deletion requests) or influence estimation to define forget sets.
> - **Retain set**: Since unlearning aims to retain knowledge of the original model outside of the data to be forgotten, an ideal retain set is very close in distribution to the forget set, yet does not contain a knowledge overlap. Unlearning benchmarks like WMDP, which we also support, consider real world unlearning of harmful content, for instance.
>
> We will update our draft to discuss the above dataset identification steps. Our framework supports a variety of methods across different unlearning setups: that optionally use a retain set (e.g., NPO) or operate without one (e.g., UNDIAL).
>
> ---
>
> We hope the above discussion this clarifies how our work fits within the broader unlearning ecosystem and complements the important identification research you've highlighted. **Please let us know if any additional analysis in particular would help strengthen your assessment of our contributions.**

---

> > ### Comment · Reviewer_QUKS · 2025-08-05
> >
> > Thank you for your response. My concerns have been addressed. Since I have give a high score, I would like the maintain this score. Thank you.

---

### Official Review · Reviewer_P4bn · 2025-07-03

**Rating:** 5
**Confidence:** 3

**Summary:**

This paper introduces OpenUnlearning, a unified, extensible benchmarking framework designed to standardize and accelerate research in unlearning for large language models (LLMs). The framework integrates 9 unlearning algorithms, 16 evaluation metrics, and 3 widely-used benchmarks (TOFU, MUSE, WMDP), aiming to address fragmentation and reproducibility challenges in the field. The authors also provide a meta-evaluation of unlearning evaluation metrics themselves, using a large pool of released LLM checkpoints, to assess their faithfulness and robustness.

**Additional Feedback:**

None

**Dataset Code Accessibility:**

Yes

**Dataset Code Comments:**

The code is accessible and with a comprehensive and well-structured README. Detailed instructions are provided to facilitate reproducibility.

**Ethical Comments:**

The authors have thoughtfully considered privacy, safety, and fairness implications of unlearning, both in the motivation and in the evaluation of unlearning effectiveness. No significant unaddressed ethical concerns remain.

**Ethical Considerations:**

No, there are no or only very minor ethics concerns

**Limitations Weaknesses:**

1.  The primary contribution is integrative and evaluative, rather than a new unlearning algorithm or metric. While the standardization is valuable, it is more incremental on the methodological side. The framework largely aggregates existing methods, and despite some technical improvements, methodological innovation is limited.
2.  The ground-truth pools for meta-evaluation are based on synthetic or semi-natural forget/retain splits. Section 4.1 describes this design, but real-world deletion requests are much more variable and adversarial. Thus, the faithfulness and robustness results might not generalize perfectly.

**Strengths Contributions:**

1.  The unification and extensibility of benchmarking for LLM unlearning is a substantial advance. OpenUnlearning brings together a fragmented field, offering a shared library that is modular and easy to extend.
2.  The integrated meta-evaluation benchmark for unlearning metrics is highly impactful. By constructing a large, diverse pool and providing concrete protocols for assessing faithfulness and robustness, the framework moves beyond qualitative arguments to quantitative, reproducible assessment of metric reliability.
3.  The inclusion of public code, released models, and tooling under open licensing, with evidence of community uptake (250+ GitHub stars), ensures practical relevance and long-term impact.
4.  All key claims are backed by well-chosen figures and tables. The language is precise, and references to prior work are apt and exhaustive.

---

> ### Author Rebuttal · Authors · 2025-07-31
>
> We sincerely thank you for your thoughtful review and for recognizing OpenUnlearning as a "substantial advance" that provides "quantitative, reproducible assessment of metric reliability." We're encouraged by your assessment of our meta-evaluation framework's "high impact" and the recognition that our work addresses fragmentation in the field.
>
> ---
>
> ## *Summary of responses*
>
> | Concern                          | Response                                                                                             |
> |----------------------------------|------------------------------------------------------------------------------------------------------|
> | Limited methodological innovation | Emphasized technical contributions: cross-benchmark implementations, meta-evaluation design with formal metric definitions. First work in literature on evaluating evaluations.    |
> | Meta-evaluation generalization    | Our evaluation with diverse unlearning methods on adversarial stress tests shows practical relevance. |
>
> ---
>
> ## 1. Response to Methodological Innovation Concerns
>
> You note that our "primary contribution is integrative and evaluative, rather than a new unlearning algorithm or metric" with "limited methodological innovation." While integration is indeed our primary contribution, we would like to point to our technical advances that go beyond simple aggregation:
>
> 1. **Meta-evaluation**: We introduce formal mathematical definitions for faithfulness and robustness of unlearning metrics, with concrete measurement protocols (Equations 1-3 in Section 4.2-4.3). We perform the first-of-its-kind benchmarking of unlearning evaluation *metrics*. As has been noted in many previous works [1,2], the problem of evaluating unlearning is as hard as unlearning itself. We believe our work makes a crucial step in this direction.
> 2. **Cross-benchmark metric unification**: We developed unified implementations enabling metrics to work across different benchmarks (e.g., MUSE's PrivLeak now works in TOFU, LM-Eval integration across all benchmarks), requiring substantial technical work to handle different data formats and evaluation protocols.
>
>
> These contributions required methodological and engineering work beyond simple aggregation of existing tools.
>
> [1] Lucki et. al 2024. An Adversarial Perspective on Machine Unlearning for AI Safety.
> [2] Schwarzschild et. al. 2024. Rethinking LLM Memorization through the Lens of Adversarial Compression.
>
> ---
>
> ## 2. Response to Meta-Evaluation Limitations
>
> You correctly note that our "ground-truth pools for meta-evaluation are based on synthetic or semi-natural forget/retain splits", and question generalization to "real-world deletion requests."
>
> - **Practicality**: While you're correct that real-world deletion requests are "more variable and adversarial," our synthetic setup enables controlled evaluation of metric properties that would be hard to assess systematically with purely real-world data. In particular, we wanted to establish a minimal set of criteria or properties that any reliable unlearning evaluation metric should satisfy.
>
> We would also like to note that our meta-evaluation setup covers a diverse range of unlearning approaches and adversarial scenarios:
>
> - **Method diversity**: Our evaluation spans 8 diverse unlearning methods (GradDiff, NPO, SimNPO, RMU, UNDIAL, AltPO, etc.), each creating different "ground truth" unlearning states through different mechanisms. With 450+ models across different hyperparameter settings and training variants, we provide a comprehensive ground-truth pool for meta-evaluation in terms of unlearning approaches.
> - **Stress-testing realism**: Our relearning attacks specifically test real-world adversarial scenarios where deleted data might be re-encountered. Our quantization tests address practical deployment concerns where models undergo compression.
>
> ---
>
> We hope this clarifies the technical depth behind our integration work and demonstrates the practical relevance of our meta-evaluation results. **Please let us know if any additional analysis in particular would help address any remaining concerns.**

---

> > ### Author Response · Authors · 2025-08-05
> >
> > Dear reviewer,
> >
> > Thank you for your thoughtful review. We believe we've addressed your concerns and would welcome any remaining feedback. If our responses have resolved the issues you raised, we'd be grateful if this could be reflected in your assessment.

---

### Decision · Program_Chairs · 2025-09-18

**Decision:**

Accept (poster)

**Comment:**

[Paper Summary]
This paper presents OpenUnlearning, a unified framework for benchmarking LLM unlearning. It integrates 9 algorithms, 16 metrics, and 3 benchmarks (TOFU, MUSE, WMDP), releasing 450+ checkpoints. A notable contribution is the meta-evaluation of unlearning metrics, defining faithfulness and robustness as core desiderata.

[Assessment of Reviews]
Most reviewers were positive, citing strong community value, reproducibility, and novelty of the meta-evaluation. Concerns centered on limited methodological novelty, reliance on synthetic forget/retain splits, and ranking sensitivity. Suggestions included adding Qwen models, RWKU benchmark, and clarifying privacy-memorization trade-offs. The rebuttal addressed issues adequately.

[Justification for Decision]
With scores of 4, 5, 5, 5, I recommend Accept. The work is timely, technically solid.

[Comment]
For the camera-ready, I suggest (1) clarify generalization beyond synthetic splits, (2) support for new models/benchmarks, and (3) discuss privacy-utility trade-offs more explicitly.